# A Mechanistic View of Catastrophic Overfitting

## Abstract

Adversarial Training (AT) suffers from a critical failure mode known as Catastrophic Overfitting (CO), where robustness to weak single-step adversaries does not translate to strong multi-step adversaries. Despite progress in mitigating CO, its underlying mechanisms remain poorly understood. In this work, we address two central questions: (1) Why does CO appear? and (2) What role do the number of Projected Gradient Descent (PGD) steps and PGD initialization play in CO? Using mathematically tractable models, we reveal a phase transition in the adversarial budget $\epsilon$, above which non-robust solutions become optimal. Furthermore, we show that CO exists for any well separated dataset, any number of PGD steps $S$, $\epsilon$ as small as desired, and randomized initialization. Our insights align with empirical observations in the community and help explain the difficulties in avoiding CO at larger scales. We believe our results deepen the understanding of CO and provide a foundation for developing future-proof solutions.

## 1 Introduction

Neural Networks (NNs) are not robust by default to small corruptions in their input (Biggio et al., 2013; Szegedy et al., 2014). Adversarial Training (AT) (Madry et al., 2018) and its variants have proven to be one of the most effective strategies to achieve robust models (Croce et al., 2020). Formulated as a min-max problem, AT can be solved via an alternating max-min procedure (Danskin, 1966; Latorre et al., 2023). However, solving the inner max problem, typically with Projected Gradient Descent (PGD), significantly slows down training, fostering the use of single-step methods (Shafahi et al., 2019; Wong et al., 2020).

Unfortunately, when solving the inner max problem with a single Fast Gradient Signed Method (FGSM) step (Goodfellow et al., 2015), AT can result in a critical failure mode known as Catastrophic Overfitting (CO) (Wong et al., 2020; Andriushchenko & Flammarion, 2020). CO generally occurs abruptly for an adversarial budget $\epsilon$ larger than a threshold $\epsilon_c$, where the model overfits to be 100% robust to (weak) single-step adversaries while being 0% robust to (strong) multi-step adversaries. For example, Andriushchenko & Flammarion (2020) observed $\epsilon_c = 6/266$ for FGSM in CIFAR10.

Recently, different solutions to CO have been explored (de Jorge et al., 2022; Sriramanan et al., 2020). Almost simultaneously, several solutions have been shown to fail (Andriushchenko & Flammarion, 2020; Abad Rocamora et al., 2024) and even mechanisms to induce CO have been proposed (Ortiz-Jimenez et al., 2023). Such developments have increased uncertainty regarding the appearance of CO, leaving the following question unanswered:

**Q1:** *Why does catastrophic overfitting appear?*

While **Q1** remains open, the community has built useful intuitions regarding the cases when CO does *not* appear, i.e., using a large number of PGD steps (Andriushchenko & Flammarion, 2020) and introducing noise in the initialization of PGD (de Jorge et al., 2022). Strategies implemented in the seminal paper of Madry et al. (2018).

Contrary to these intuitions, He et al. (2023) demonstrated that CO can occur in multi-step AT with large step-sizes and Abad Rocamora et al. (2024) showed that noise-based methods present CO for larger $\epsilon$ or

---

**Algorithm 1** PGD Adversarial Training (AT) (Madry et al., 2018).

---

1: **Inputs:** Model weights $\boldsymbol{\theta}$, Dataset $\{\boldsymbol{x}_i, y_i\}_{i=1}^n$, # epochs $T$, # batches $M$, radius $\epsilon$, PGD step-sizes $\alpha_s$, learning rate $\gamma$ and initialization radius $\sigma$.

2: **for** $t \in [T]$ **do**

3:      **for** $i \in [M]$ **do**

4:          $\boldsymbol{\delta}_0^i = \text{Unif.}\left([-\sigma, \sigma]^d\right)$                                              ▷ Initialize perturbation

5:          **for** $s \in [S]$ **do**                                                  ▷ Solving inner max

6:              $\boldsymbol{\delta}_s^i = \boldsymbol{\delta}_{s-1}^i + \alpha_s \cdot \epsilon \cdot \text{sign}\left(\nabla_{\boldsymbol{x}_i} \mathcal{L}(\boldsymbol{f_\theta}(\boldsymbol{x}_i + \boldsymbol{\delta}_{s-1}^i), y_i)\right)$      ▷ Signed gradient ascent step

7:              $\boldsymbol{\delta}_s^i = \min\{\max\{\boldsymbol{\delta}_s^i, -\epsilon\}, \epsilon\}$                       ▷ Project so that $\left|\left|\boldsymbol{\delta}_s^i\right|\right|_\infty \leq \epsilon$

8:          $\boldsymbol{\theta} = \boldsymbol{\theta} - \gamma \cdot \nabla_{\boldsymbol{\theta}} \mathcal{L}(\boldsymbol{f_\theta}(\boldsymbol{x}_i + \boldsymbol{\delta}_S^i), y_i)$                               ▷ SGD update

---

larger model sizes. These contradictory results suggest that the aforementioned strategies do not truly avoid CO on their own for any adversarial budget, and pose the following question:

> **Q2:** *What is the role of the number of PGD steps $S$ and its initialization in the appearance of CO?*

In this work, we provide a broader understanding of **Q1** and **Q2**. Regarding **Q1**, we craft mathematically tractable models, where we can demonstrate the existence of a phase transition in $\epsilon$, which above $\epsilon > \epsilon_c$ results in a non-robust solution being the optimal classifier for the AT problem. We are also able to show a negative result regarding **Q2**: CO exists for any well-separated dataset, and any $S$, with and without random initialization. Our insights agree with empirical observations in large scale NNs and datasets by the community. Overall, our paper provides a mechanistic understanding of catastrophic overfitting. We believe that incorporating our insights, the community will be fostered to develop future-proof solutions to CO.

The rest of the paper is organized as follows: In Sec. 2, we set up the problem of CO in AT, as well as the current paradigms concerning origins and solutions. In Sec. 3 we introduce two different models where CO manifests as a phase transition. In Secs. 4 and 5, we consider multiple step attacks and random PGD initialization. We confirm that the insights gained from the theory are observed in large scale, real-world settings in Sec. 6, and conclude in Sec. 7.

**Notation:** We use uppercase bold letters for matrices $\boldsymbol{X} \in \mathbb{R}^{m \times n}$, lowercase bold letters for vectors $\boldsymbol{x} \in \mathbb{R}^m$ and lowercase letters for numbers $x \in \mathbb{R}$. Accordingly, the $i^{\text{th}}$ row and the element in the $i, j$ position of a matrix $\boldsymbol{X}$ are given by $\boldsymbol{x}_i$ and $x_{ij}$ respectively. We use the shorthand $[n] = \{1, \cdots, n\}$ for any natural number $n$. We denote the indicator function as $\mathbb{1}[\cdot]$.

## 2 Background

In Sec. 2.1 we analyze AT. Then, in Sec. 2.2 we cover single-step AT and the CO problem. Next, in Sec. 2.3 we cover the efforts towards avoiding CO. Finally, in Sec. 2.4 we cover the existing explanations of the CO phenomenon.

### 2.1 Adversarial training

AT can be formulated as a min-max optimization problem. Let $\{(\boldsymbol{x}_i, y_i)\}_{i=1}^n$ be the training dataset, with $\boldsymbol{x}_i \in \mathbb{R}^d$ and $y_i \in [o]$. Let $\boldsymbol{f_\theta} : \mathbb{R}^d \to \mathbb{R}^o$ be a classifier parameterized by $\boldsymbol{\theta} \in \mathbb{R}^p$, assigning a score to each class so that the predicted class is given by $\hat{y}_i = \arg\max_{j \in [o]} \boldsymbol{f_\theta}(\boldsymbol{x}_i)_j$. Let $\mathcal{L} : \mathbb{R}^o \times [o] \to \mathbb{R}^+$ be a loss function, the adversarial training problem can be formulated as:

$$\min_{\boldsymbol{\theta} \in \mathbb{R}^p} \frac{1}{n} \sum_{i=1}^n \max_{||\boldsymbol{\delta}_i||_\infty \leq \epsilon} \mathcal{L}\left(\boldsymbol{f_\theta}(\boldsymbol{x}_i + \boldsymbol{\delta}_i), y_i\right). \tag{AT}$$

Danskin's theorem (Danskin, 1966) allows solving Eq. (AT) as a minimization problem using first order methods, where at each iteration, the gradient is computed by (approximately) solving the inner maximization problem of Eq. (AT). In order to solve the inner max, PGD is commonly used (Madry et al., 2018). The standard AT procedure is covered in Alg. 1.[1]

The main drawback of AT is that solving the inner maximization problem considerably slows down training in comparison to standard training. This has motivated the use of single-step ($S = 1$ in Alg. 1) AT variants (Shafahi et al., 2019; Wong et al., 2020; Andriushchenko & Flammarion, 2020).

## 2.2 Single-step AT and Catastrophic Overfitting

In order to alleviate the overhead arising from solving the inner maximization problem, Shafahi et al. (2019) propose Free-AT: computing the gradients with respect to the weights and perturbations in parallel, in order to update them simultaneously for some iterations over the same batch. This is equivalent to moving line 8 in Alg. 1 inside the PGD loop. Alternatively, a single-step (FGSM) attack (Goodfellow et al., 2015) can be employed. This is equivalent to setting $S = 1$ and $\alpha = 1$ in Alg. 1.

Unfortunately, Free-AT and FGSM training often result in the so called *Catastrophic Overfitting* (CO) phenomenon (Wong et al., 2020).[2] We define CO as follows:

**Definition 2.1** (($\beta, \eta$)-Catastrophic Overfitting (CO)). Let $\beta, \eta \in [0, 1]$. Let $\boldsymbol{f_\theta} : \mathbb{R}^d \to \mathbb{R}^o$ be a classifier trained with Alg. 1 on a dataset $\{(\boldsymbol{x}_i, y_i)\}_{i=1}^n$ and adversarial budget $\epsilon$. The ($\beta, \eta$)-CO phenomenon is characterized by a high PGD accuracy and low robust accuracy:

$$\frac{1}{n} \sum_{i=1}^n \mathbb{1} \left[ \arg\max_{j \in [o]} f_{\boldsymbol{\theta}}(\boldsymbol{x_i} + \boldsymbol{\delta}_S^i)_j = y_i \right] \geq 1 - \beta, \quad \frac{1}{n} \sum_{i=1}^n \mathbb{1} \left[ \arg\max_{j \in [o]} f_{\boldsymbol{\theta}}(\boldsymbol{x_i} + \boldsymbol{\delta}_\star^i)_j = y_i \right] \leq \eta,$$

for some $\boldsymbol{\delta}_\star^i : \left\| \boldsymbol{\delta}_\star^i \right\|_\infty \leq \epsilon \quad \forall i \in [n]$ and $\boldsymbol{\delta}_S^i$ obtained with the same PGD hyperparameters as used for training with Alg. 1. Intuitively, CO results in a model being robust to the PGD attacks seen during training, but not being robust to other $\ell_\infty$ perturbations bounded by $\epsilon$. In the remaining of this work, we suppress the ($\beta, \eta$) prefix unless needed.

CO is characterized for appearing abruptly for $\epsilon$ values above a threshold $\epsilon_c$ (Andriushchenko & Flammarion, 2020), defined as:

**Definition 2.2** (Critical adversarial budget $\epsilon_c$). Let $\beta, \eta \in [0, 1]$. Let $\boldsymbol{f_\theta} : \mathbb{R}^d \to \mathbb{R}^o$ be a classifier trained with Alg. 1 on a dataset $\{(\boldsymbol{x}_i, y_i)\}_{i=1}^n$. For a given set of training hyperparameters, $\epsilon_c$ is the critical adversarial budget, if ($\beta, \eta$)-CO is observed for all adversarial budgets $\epsilon > \epsilon_c$, but ($\beta, \eta$)-CO is not observed for any $\epsilon \leq \epsilon_c$.

## 2.3 Current solutions

In order for the FGSM solution of the inner max to be accurate, the loss needs to be *locally linear* (Andriushchenko & Flammarion, 2020):

$$\mathcal{L} \left( \boldsymbol{f_\theta}(\boldsymbol{x}_i + \boldsymbol{\delta}^i), y_i \right) \approx \mathcal{L} \left( \boldsymbol{f_\theta}(\boldsymbol{x}_i), y_i \right) + \boldsymbol{\delta}^{i\top} \boldsymbol{g}_i, \tag{1}$$

where $\boldsymbol{g}_i = \nabla_{\boldsymbol{x}_i} \mathcal{L} \left( \boldsymbol{f_\theta}(\boldsymbol{x}_i), y_i \right)$ for all $(\boldsymbol{x}_i, y_i)$ in the dataset and $\boldsymbol{\delta}^i : \left\| \boldsymbol{\delta}^i \right\|_\infty \leq \epsilon$. Crucially, this property is lost during training when CO appears. It was initially thought that CO could be solved by adding uniform noise to the input to add diversity in the gradient estimation (Tramèr et al., 2018; Wong et al., 2020; Kang & Moosavi-Dezfooli, 2021; de Jorge et al., 2022). However, it was later shown that this only postpones the appearance of CO to larger $\epsilon$ (Andriushchenko & Flammarion, 2020; Abad Rocamora et al., 2024). In order to explicitly enforce Eq. (1) during single-step AT, multiple approaches have been proposed to regularize

---

[1]Note that in Line 6 of Alg. 1, taking the gradient with respect to the perturbation $\boldsymbol{\delta}_{s-1}^i$ or the input $\boldsymbol{x}_i$ is equivalent, i.e., $\nabla_{\boldsymbol{x}_i} \mathcal{L}(\boldsymbol{f_\theta}(\boldsymbol{x}_i + \boldsymbol{\delta}_{s-1}^i), y_i) = \nabla_{\boldsymbol{\delta}_{s-1}^i} \mathcal{L}(\boldsymbol{f_\theta}(\boldsymbol{x}_i + \boldsymbol{\delta}_{s-1}^i), y_i)$.

[2]Not to be confused with the taxonomy of Mallinar et al. (2022) for generalization in overparametrized neural networks.

quantities related to the local linearity error (Moosavi-Dezfooli et al., 2019; Qin et al., 2019; Andriushchenko & Flammarion, 2020; Abad Rocamora et al., 2024).

Other approaches for avoiding CO involve regularizing the norm of the difference in logits between clean and adversarial samples (Sriramanan et al., 2020; 2021), performing AT in the latent space (Park & Lee, 2021), constraining the weights to be in a subspace during AT (Li et al., 2022), reformulating AT as a bilevel problem (Zhang et al., 2022), zeroing small adversarial perturbations (Golgooni et al., 2023), reducing the number of abnormal adversarial examples (Lin et al., 2023), modifying the initialization of PGD (Jia et al., 2022; 2024; Pan et al., 2024), or combining AT with Sharpness Aware Minimization (Foret et al., 2021; Lin et al., 2024).

### 2.4 Current explanations

**The appearance of CO:** Several works have tried to explain why CO occurs. Kim et al. (2021) argue the loss landscape becomes "distorted" when CO appears, meaning that the original point $x$ and the FGSM adversarial example $x_{\mathrm{FGSM}}$ are well classified. However, there exist misclassified points closer to $x$, even in the convex combination of $x$ and $x_{\mathrm{FGSM}}$. He et al. (2023) analyze CO from the "self fitting" perspective and argue that during FGSM training, the network encodes features into the gradient, relying less on the features of the data itself for classification and resulting in CO. Recently, Pan et al. (2025) similarly observed that label information from certain samples transfers across samples. Ortiz-Jimenez et al. (2023) show that CO can be induced by discriminative features that are relevant for classification, but not robustness. Concretely, for every training sample $(x_i, y_i)$, a label-dependent mask can be added to the input $\hat{x}_i = x_i + \beta \cdot v(y_i)$. When training with the modified dataset using certain $\beta$ values, CO can be induced even for adversarial budgets as small as $\epsilon = 4/255$ in CIFAR10.

**CO in multi-step AT:** According to Andriushchenko & Flammarion (2020), multi-step AT implicitly enforces local linearity (Eq. (1)) and avoids CO. Conversely, He et al. (2023) show that CO can happen in multi-step AT with large step sizes ($\alpha$).

Although we understand the typical model behavior when CO occurs, can induce CO, and even find ways to avoid it, existing literature fails to provide a reason for the appearance of CO. Moreover, little is known about the relationship between the number of PGD steps, PGD initialization and CO. In this work, we contribute to a broader understanding of these points by addressing **Q1** and **Q2**.

## 3 Catastrophic Overfitting as a phase transition

In this section, we construct a simple model to reproduce CO. Studying this model allows us to analyze the phase transition resulting in CO, providing a better understanding of **Q1**. In Sec. A we provide additional details for this model and propose a model with a ReLU NN, where phase transitions are also observed.

### 3.1 Toy model

We consider a training dataset consisting of two points $x_1 = -x_2 \in \mathbb{R}$, with $x_1 = -\pi/2$. The task is binary classification, with labels generated by taking $y = \Theta(x)$, where $\Theta(z)$ is the Heaviside function. We use the cross-entropy (CE) loss $\mathcal{L}(\theta, \epsilon) = \frac{1}{2} \sum_{i=1}^{2} \mathcal{L}(f_\theta(x_i), y_i)$, where $\mathcal{L}(f_\theta(x_i), y_i) = -y_i \log p_i - (1 - y_i) \log(1 - p_i)$, and the probabilities are simply $p_i = e^{f_\theta(x_i)}/(1 + e^{f_\theta(x_i)})$. We take the classification NN to be:

$$f_\theta(x_i) = \sin(\theta x_i), \qquad \theta \in [0, \theta_{\max}], \tag{2}$$

where $\theta$ is the single NN parameter. Note that the NN weight is bounded by $\theta_{\max}$, which we take to be $\theta_{\max} = 10$ for the rest of this section. This is done to match real-world settings in which weights do not diverge, while at the same time consisting a large enough value interval to observe CO. The remainder of this section elaborates on the behavior of Eq. (2). The loss optimized by Alg. 1 with $S = 1$ and $\alpha = 1$, is given by:

$$\mathcal{L}(\theta, \epsilon) = \mathcal{L}\left(f_\theta\left(x_i + \epsilon \cdot \mathrm{sign}\left(\frac{d\mathcal{L}(f_\theta(x_i), y_i)}{dx_i}\right)\right), y_i\right) = \log\left(1 + e^{-\sin\left(\theta\left(\frac{\pi}{2} - \epsilon\,\mathrm{sign}\left(\theta\cos\left(\frac{\theta\pi}{2}\right)\right)\right)\right)}\right). \tag{3}$$

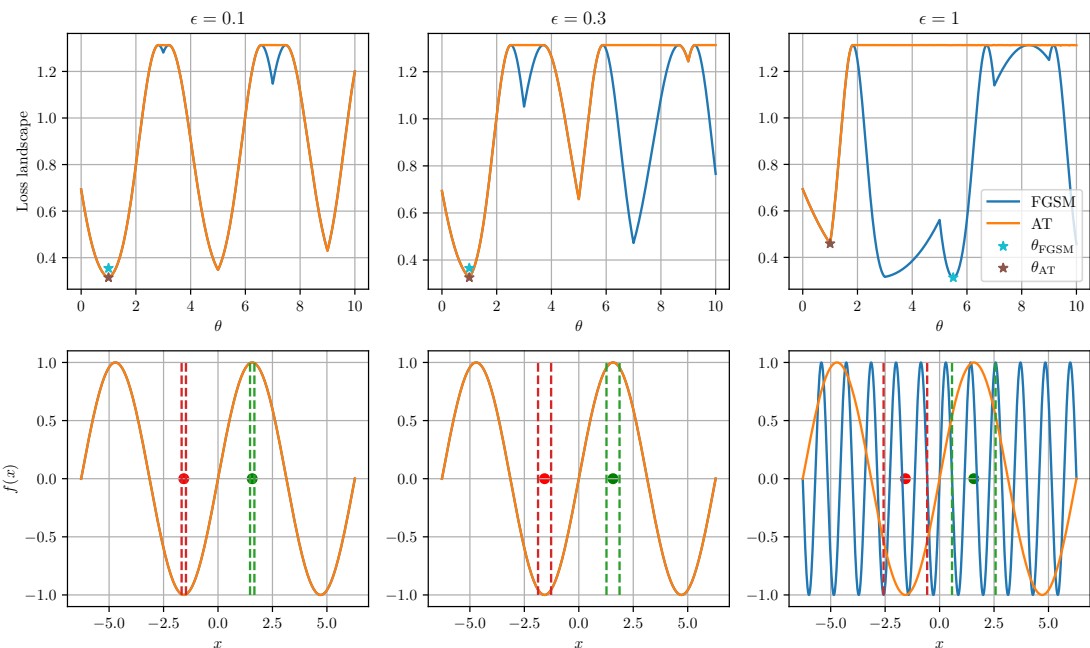

Figure 1: **Loss landscape and optimal classifiers for the toy model and** $\epsilon \in \{0.1, 0.3, 1\}$**:** In the *top row*, we plot the loss function with respect to the single parameter in the model $\theta$. In the *bottom row*, we plot the optimal classifiers according to the loss in the upper row, the two training points (● and ●) and their adversarial regions (‑‑ and ‑‑). The optimal classifier obtained with FGSM (★) coincides with one obtained with AT (★) for $\epsilon \in \{0.1, 0.3\}$. For $\epsilon = 1 > \epsilon_c$, the FGSM and AT solutions differ, with FGSM obtaining distorted solutions where the FGSM-attacked points are well classified, but clean samples are not.

In Fig. 1 we visualize Eq. (3) (—) and the true AT loss in Eq. (AT) (—) for $\epsilon \in \{0.1, 0.3, 1\}$ and $\theta \in [0, \theta_{\max}]$. Visually, for the small $\epsilon \in \{0.1, 0.3\}$ values, the robust minimum ($\theta = 1$) is the solution for both the FGSM and AT objectives. For the large $\epsilon = 1$, a larger (non-robust) value of $\theta$ attains a smaller FGSM loss than $\theta = 1$, but a very high robust loss. This change of minimums is characteristic of a phase transition on $\epsilon$. Next, we analyze this phase transition in detail.

## 3.2   Phase transition in the toy model

The loss in Eq. (3) exhibits interesting properties when varying the perturbation $\epsilon$. When $\epsilon = 0$, the loss has infinitely many degenerate minima, at $\theta_{\min} = 4n + 1$, where $n \in \mathbb{Z}^+$. While these minima share the same loss values, they differ in loss curvature with respect to the samples $x_i$, measured by the sample-wise Hessian $H(\theta) \equiv \frac{d^2}{dx_i^2} \mathcal{L}(f_\theta(x_i), y_i)|_{x_i = \pm\frac{\pi}{2}}$, given by:

$$H(\theta) = \frac{\theta^2 \left( \sin\left(\frac{\pi\theta}{2}\right) + e^{\sin\left(\frac{\pi\theta}{2}\right)} \left( \sin\left(\frac{\pi\theta}{2}\right) + \cos^2\left(\frac{\pi\theta}{2}\right) \right) \right)}{2 \left( 1 + e^{\sin\left(\frac{\pi\theta}{2}\right)} \right)^2} \propto \theta^2, \tag{4}$$

which increases with $\theta$, implying that the optimal solution with the lowest curvature solution is at $\theta_{\min} = 1$, which is indeed the solution for $\epsilon \to 0$. However, as $\epsilon$ is increases, the loss landscape changes, and certain minima obtain higher or lower loss values, changing both the local and global minima. In particular, at $\epsilon_c = \pi/8$, the minimum at $\theta = 1$ becomes unstable, and the loss is instead minimized at $\theta = 7$, which was originally a maximum of the loss. We explicitly show that this occurs for the toy model in Fig. 2.

The abrupt transition of the system from one minimum to another is precisely the behavior observed in physical systems which undergo a first order phase transition, whose theory is well understood (Landau & Lifshits, 1958). These transitions exhibit a discontinuity in the first derivative of the free energy with

respect to some thermodynamic variable, analogized here by the discontinuity of the derivative of the loss with respect to $\theta$ at the transition point $\epsilon_c$. We show how the loss landscape evolves with $\epsilon$ in Fig. 1, as well as the solutions found by the network $f(x)$. Clearly, for small $\epsilon < \epsilon_c$, the system is in a "generalizing" phase, where the curvature is low and the optimal robust solution is obtained. However, for $\epsilon \geq \epsilon_c$, the model is minimized at an "overfitting" phase, where the perturbed data points are overfit by favoring a high curvature solution, completely failing on the original data. This example illustrates that the essence of CO lies in the phase transition from a low curvature to a high curvature region of the solution.

In practice, since the toy model depends on only one variable, a grid search can be done to solve the AT problem. We can then scan over $\epsilon$ to empirically find $\epsilon_c$. In Fig. 2, we did precisely this and measured the FGSM (−) and AT (−) objectives for $\theta_{\text{FGSM}}$ and a wide range of values $\epsilon \in [0, 1]$, finding that the point where the phase transition occurs and the two losses diverge to be $\epsilon_c = \pi/8$. For additional experimental details we refer to Sec. 6.

Overall, the analysis of CO as a phase transition sheds light on **Q1**. Our theoretical results in this section, jointly with the exact same phenomenon occurring at larger-scale and more complex models (Andriushchenko & Flammarion, 2020), suggest that similar phase transitions occur when training deep neural networks on large scale datasets. We refer to Sec. 6.2 for an empirical analysis in the MNIST, SVHN and CIFAR10 datasets.

Figure 2: **Phase transition in the toy model:** The critical value is $\epsilon_c = \pi/8$. We evaluate the FGSM and AT losses at the optimal weight for the FGSM loss, i.e., $\theta_{\text{FGSM}}$. For $\epsilon > \epsilon_c$, the phase transition occurs, resulting in a sudden increase in the AT loss and a decrease in the FGSM loss.

## 4 CO in multi-step AT

The current intuition is that multi-step AT avoids CO and preserves local linearity (Moosavi-Dezfooli et al., 2019). Nevertheless, examples where CO appears under a small number of steps (Andriushchenko & Flammarion, 2020) or a large stepsize (He et al., 2023) have been found. In this section we address these points and answer **Q2**.

In order to guarantee that CO cannot appear, we need to show that no CO solution exists in the loss landscape. Similarly to our analysis in Sec. 3, we are able to show that unfortunately, CO solutions exist in the multi-step setting, agreeing with the empirical findings of He et al. (2023).

In Thms 4.1 and 4.2, we present an example where CO appears for arbitrarily small $\epsilon$ and arbitrarily large $S$. We then analyze the interplay between the norm of the model parameters $|\theta|$ and the number of steps $S$ leading to CO.

**Theorem 4.1** (Solutions of the PGD AT problem in the toy model $(\sigma = 0)$)**.** *Let the classifier defined in Sec. 3.1 be trained with Alg. 1, $S$ steps, PGD step sizes $\alpha_s = 1/S$, $\forall s = 1, \cdots, S$ and $\sigma = 0$. Let $a > 0$ and $b_k = \frac{1+4\cdot k}{1-\frac{1}{a}}$ for $k = 1, \cdots, \infty$, the pair $\theta_k = b_k$, $\epsilon_k = \frac{2\pi S}{b_k}$, gives us weights $\theta_k$ for the corresponding adversarial budget $\epsilon_k$, so that:*

$$\frac{1}{2} \sum_{i=1}^{2} \mathcal{L}(f_{\theta_k}(x_i + \delta_S^i), y_i) - \mathcal{L}^\star \leq \pi \frac{b_k}{a} \,,$$

*where $\mathcal{L}^\star = \log(1 + e) - 1$ is the optimal loss for a classifier defined as in Sec. 3.1.*

**Corollary 4.2** $((\beta = 0, \eta = 0)$-CO exists at arbitrarily small $\epsilon$ for any $S$)**.** *Given any $S$, by increasing $a$ and $k$ we can take $\epsilon_k$ arbitrarily close to zero with arbitrarily accurate solutions $\theta_k$, where the points $x_i \pm \frac{\epsilon_k}{2S}$ are misclassified.*

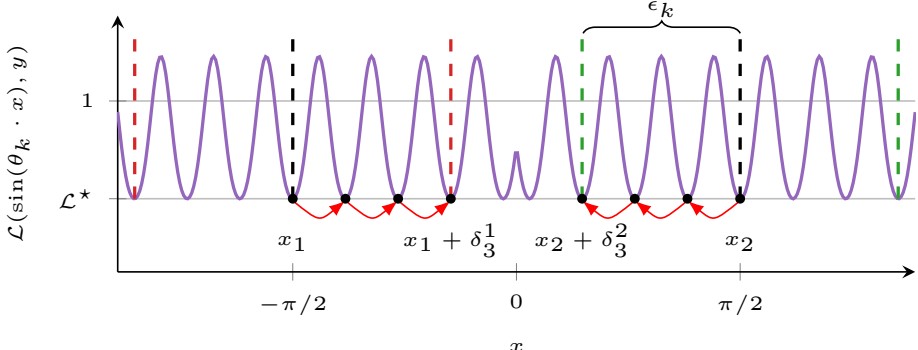

Figure 3: **CO exists in multi-step AT for arbitrarily small $\epsilon$:** Visualization of Thm 4.1 for $S = 3, a = 1,000$ and $k = 4$. We can construct an optimal classifier where PGD converges to a point with low loss, while the true max of the loss is much higher.

**Corollary 4.3** (Bounding $|\theta|$ and increasing $S$ can avoid CO). *Let $b_k$ be constrained as $|b_k| \leq B$, we have that $\epsilon_k \geq \frac{2\pi S}{B}$. Meaning that bounding the norm of $\theta$ and increasing the number of steps can help avoid CO by avoiding the solutions in Thm 4.1.*

*Remark* 4.4 (Additional CO solutions). In Thm 4.1 we present sufficient conditions to observe CO. Nevertheless, other solutions not captured by Thm 4.1 can be observed, see Fig. 1

In Thm 4.1, we provide a constructive mechanism to build PGD AT solutions where CO appears for arbitrarily small $\epsilon$ Thm 4.2. This is depicted in Fig. 3, where for $S = 3$, $a = 1,000$, setting $k = 4$ results in $\epsilon_k \approx 1.1077$ and the solution $\theta_k \approx 16.98301$, for which both 3-step PGD points $(x_i + \delta_3^i)$ are well classified, but the true adversarial accuracy is zero as shown in Thm 4.2.

In Thm 4.3, we show that by looking at the ratio defining $\epsilon_k = \frac{2\pi S}{\theta_k}$, it is clear that bounding the norm of the parameters can lowerbound $\epsilon_k$ and increasing $S$ will improve such lowerbound. This avoids the solutions constructed in Thm 4.1 and, more generally, helps mitigate CO. In Sec. 6.2, we observe a similar phenomenon in real world datasets and models. Overall, regarding **Q2**, we can conclude that increasing $S$ only helps under the condition that the complexity of the model is constrained, e.g., bounding the norm of the parameters.

## 5 The influence of noise in CO

In Sec. 4 we were able to provide a model where PGD AT presents CO for any given $S$ and $\sigma = 0$. Intuitively, using $\sigma = 0$ means the PGD attack is always initialized at the same point. Then, *is this the reason behind CO? Can we avoid CO by initializing the PGD attack uniformly in the $\ell_\infty$ ball?*

In Sec. 5.1 we show that this is not the case, we construct a simple example with a ReLU NN where CO appears for the standard $\sigma = \epsilon$ (Madry et al., 2018; Wong et al., 2020). Then, in Sec. 5.2, we extend the model to any well separated dataset.

### 5.1 A simple model presenting CO with $\sigma = \epsilon$

Let $(x_1, y_1) = (-\pi/2, 0)$ and $(x_2, y_2) = (\pi/2, 1)$ as in Sec. 3, our classifier in this section will be given by:

$$f_{\boldsymbol{\theta}}(x) = \boldsymbol{w}^{(2)\top} \mathbf{ReLU}(x\boldsymbol{w}^{(1)} + \boldsymbol{b}^{(1)}) + b^{(2)}, \tag{5}$$

where $\boldsymbol{w}^{(1)}, \boldsymbol{w}^{(2)}$ and $\boldsymbol{b}^{(1)}$ are vectors in $\mathbb{R}^6$ and $b^{(2)} \in \mathbb{R}$, $\boldsymbol{\theta} = \{\boldsymbol{w}^{(1)}, \boldsymbol{b}^{(1)}, \boldsymbol{w}^{(2)}, b^{(2)}\}$ and $\mathrm{ReLU}(x) = \max\{0, x\}$ is applied elementwise. In order to further simplify the model and the analysis, we make the model dependent on two parameters $a, b \in \mathbb{R}$:

$$
\begin{aligned}
\boldsymbol{w}^{(1)} &= \begin{pmatrix} 1 & -1 & 1 & -1 & 1 & -1 \end{pmatrix}^\top, \quad \boldsymbol{b}^{(1)} = \begin{pmatrix} 0 & 0 & -\pi/2 + a & -\pi/2 + a & -\pi/2 & -\pi/2 \end{pmatrix}^\top \\
\boldsymbol{w}^{(2)} &= \begin{pmatrix} -1 & 1 & 1 + b/a & -1 - b/a & -1 - b/a & 1 + b/a \end{pmatrix}^\top, \quad b^{(2)} = 0,
\end{aligned}
\tag{6}
$$

which simplifies to: $f_{\boldsymbol{\theta}}(x) = h(x; x_2, a, b, \pi/2) - h(-x; x_2, a, b, \pi/2)$, where

$$h(x; \mu, a, b, c) = -\text{ReLU}(x + c - \mu) + \left(1 + \frac{b}{a}\right)\text{ReLU}(x - \mu + a) - \left(1 + \frac{b}{a}\right)\text{ReLU}(x - \mu). \qquad (7)$$

Finally, we can characterize the appearance of CO with respect to the parameters in Alg. 1, $a$ and $b$.

**Theorem 5.1** (Solutions of the PGD AT problem ($\sigma = \epsilon$)). *Let $f_{\boldsymbol{\theta}}$ be a binary classifier defined as in Eq. (5) and parametrized by Eq. (6). Let the model be trained with Alg. 1, $S$ steps, PGD step sizes $\alpha_s = 1/S$ or $\alpha_s = 2/S$, $\forall s = 1, \cdots, S$ and $\sigma = \epsilon \le \pi/2$. We have that for $a \in [0, \epsilon/S)\}$):*

$$\frac{1}{2}\sum_{i=1}^{2} \mathbb{E}_{\delta_0^i \sim Unif.([-\epsilon,\epsilon])} \left[\mathcal{L}(f_{\boldsymbol{\theta}}(x_i + \delta_S^i), y_i)\right] \le e^{\pi/2} \cdot \left(e^{\epsilon-b} + \frac{S \cdot a}{2 \cdot \epsilon}\right),$$

*meaning that taking $a \to 0$ and $b \to \infty$ results in a loss arbitrarily close to zero.*

**Corollary 5.2** (CO exists with multi-step PGD AT with random initialization). *Given any $\epsilon$ and $S$, by taking $a \to 0$ and $b \to \infty$, we obtain in expectation nearly zero loss and nearly perfect accuracy for the PGD points, but the points $x_1 + a$ and $x_2 - a$ are misclassified.*

An example depicting Thms 5.1 and 5.2 is available in Fig. 4, where we can observe that the PGD trajectories have a high probability of finishing in a point with low loss.

## 5.2 Extending to any dataset

In Sec. 5.1, we observed that the important property for inducing CO with $\sigma = \epsilon$, is that the PGD trajectory falls in a low loss region with high probability. Here, we extend these findings to show the existence of CO for any number of samples $n$, input dimension $d$ and classes $o$.

Intuitively, we can construct a function that around every training point has the same behavior observed in Fig. 4. To do so, given a sample $(\boldsymbol{x}_i, y_i)$, we can make the output for the label $y_i$ be Eq. (7) along the first dimension and constant along the remaining $d-1$. Then, we can center this function in the region $\boldsymbol{x} : \|\boldsymbol{x} - \boldsymbol{x}_i\|_\infty \le \epsilon$ and sum across all the samples in the dataset. Overall, we define $\boldsymbol{f_\theta} : \mathbb{R}^d \to \mathbb{R}^o$ with $\boldsymbol{\theta} = \{a, b\}$ as:

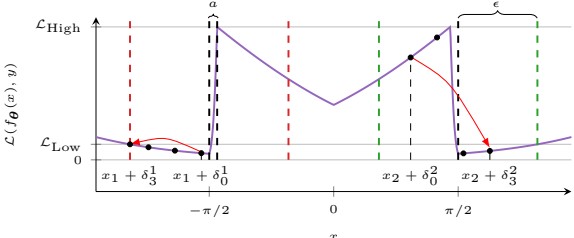

Figure 4: **Visualization of Thm 5.1 for $S = 3$ and $\epsilon = 1$:** We fix $a = 0.1$ and $b = 4$. With probability $1 - \frac{S \cdot a}{2 \cdot \epsilon}$ the PGD trajectory leads to a loss smaller than $\mathcal{L}_{\text{Low}}$, which can be taken arbitrarily close to zero by increasing $b$. This results in CO, as the PGD trajectory converges with high probability to a well classified point, but the points $x_1 + a$ and $x_2 - a$ are not well classified.

$$\boldsymbol{f_\theta}(\boldsymbol{z}) = \sum_{i=1}^{n} \boldsymbol{e}_{y_i} \cdot h(z_1; x_{i1}, a, b, \epsilon) \cdot \mathbb{1}\left[\|\boldsymbol{x}_i - \boldsymbol{z}\|_\infty \le \epsilon\right], \qquad (8)$$

where $\boldsymbol{e}_j$ is the $j^{\text{th}}$ vector of the canonical basis. It is possible to show that this classifier is optimal, leading us to the following Theorem:

**Theorem 5.3** (Solutions of the PGD AT problem for any well separated dataset ($\sigma = \epsilon$)). *Let the dataset $\{(\boldsymbol{x}_i, y_i)\}_{i=1}^{n}$ be well separated so that $\min_{i,j} \|\boldsymbol{x}_i - \boldsymbol{x}_j\|_\infty \ge 2 \cdot \epsilon$. Let the model defined in Eq. (8) be trained with Alg. 1, $S$ steps, PGD step sizes $\alpha_s = 1/S$, or $\alpha_s = 2/S$, $\forall s = 1, \cdots, S$ and $\sigma = \epsilon$. We have that for $a \in [0, \epsilon/S)\}$):*

$$\frac{1}{n}\sum_{i=1}^{n} \mathbb{E}_{\boldsymbol{\delta}_0^i \sim Unif.([-\epsilon,\epsilon]^d)} \left[\mathcal{L}(\boldsymbol{f_\theta}(\boldsymbol{x}_i + \boldsymbol{\delta}_S^i), y_i)\right] \le (o-1) \cdot e^{2 \cdot \epsilon} \cdot \left(e^{-b} + \frac{S \cdot a}{2 \cdot \epsilon}\right),$$

*meaning that taking $a \to 0$ and $b \to \infty$ results in a loss arbitrarily close to zero.*

**Corollary 5.4** (CO exists for any well separated dataset with multi-step PGD AT with random initialization). *Given any $\epsilon \le \frac{1}{2} \cdot \min_{i,j} \|\boldsymbol{x}_i - \boldsymbol{x}_j\|_\infty$ and $S$, by taking $a \to 0$ and $b \to \infty$, we obtain in expectation nearly zero loss and nearly perfect accuracy for the PGD points, but the points $\boldsymbol{x}_i - a$ $\forall i \in [n]$ are misclassified.*

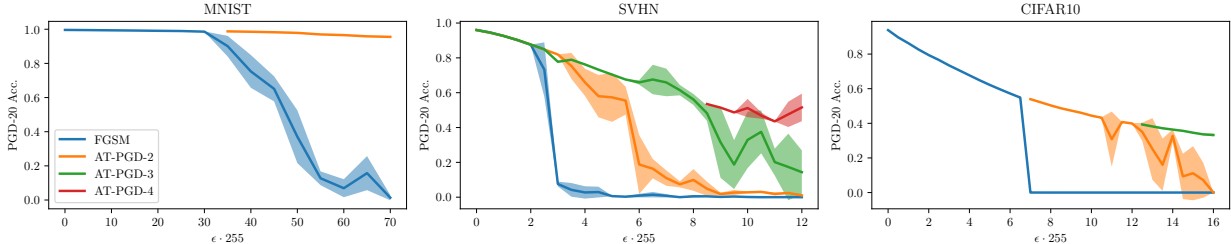

Figure 5: **The phase transition in image classification:** We train PreActResNet18 with $1, 2, 3$ and $4$ PGD steps from the first $\epsilon$ value where CO appears for one PGD step less. Larger $\epsilon$ values require more and more steps to not present CO.

The results in this section provide an explanation on why CO appears in the noisy setup employed in Wong et al. (2020) as shown by Andriushchenko & Flammarion (2020). More importantly, we show that CO solutions exist for any well separated dataset. This means that if we have a model powerful enough to approximate those solutions, which is the case for NNs (Cybenko, 1989), we cannot guarantee that Alg. 1 does not present CO. This finding agrees with the experimental results of Abad Rocamora et al. (2024), where larger ResNets, with more expressive power, presented CO when smaller ResNets did not. Answering **Q2**, we conclude that adding random initialization cannot avoid CO even if the number of steps ($S$) is high.

## 6 Experiments

In Sec. C.1 we present our experimental setup. In Sec. 6.1 we numerically analyze the loss landscape of AT and FGSM on the toy model, characterizing their global minima. Next, in Sec. 6.2, we demonstrate the phase transition ocurs in image classification datasets and models. Finally, in Sec. 6.3 we demonstrate that limiting the complexity of the classifier with weight decay can avoid CO.

### 6.1 Catastrophic overfitting in the toy model

For our toy model (Sec. 3.1), we can obtain a closed form expression for the effective loss function which is optimized by single-step AT with respect to $\theta$, see Eq. (3). Unfortunately, in order to compute the AT landscape, we would have to exactly compute for every $\theta$:

$$\delta_\star^i = \underset{\delta_i \in [-\epsilon, \epsilon]}{\arg\max} \ \mathcal{L}(\sin(\theta \cdot (x_i + \theta_i)), y_i).$$ (9)

As an alternative, we evaluate the loss at $10,000$ evenly distributed $\theta$ values in the $[0, 10]$ interval. For every $\theta$ and training sample $(x_i, y_i)$, we evaluate the loss at $100$ evenly distributed $\theta_i$ values in the $[-\epsilon, \epsilon]$ interval, and take the maximum over those in order to estimate Eq. (9). In Fig. 1 we can observe that the FGSM solution for $\epsilon = 1$ displays a distorted landscape, coinciding with the findings of Kim et al. (2021) for larger networks and datasets. In order to compute the critical value $\epsilon_c$, we repeat the procedure for $100$ evenly spaced values of $\epsilon$ in the $[0, 1]$ interval.

In Fig. 2, we report the FGSM and AT losses evaluated at the FGSM solution ($\theta_{\text{FGSM}}$). We find $\epsilon_c \approx 0.3927$, above this value, the AT loss suddenly increases and the FGSM loss starts decreasing. The latter should not be observed as the true adversarial loss monotonically increases with $\epsilon$.

### 6.2 The phase transition at a larger scale

We train PreActResNet18 (He et al., 2016) with the ReLU activation function with Alg. 1, $S \in \{1, 2, 3, 4\}$ and $\alpha_s = \epsilon/S$. We scan over $\epsilon$ values from 0 to $70/255$, $12/255$ and $16/255$ for MNIST, SVHN and CIFAR10 respectively. For each $S$, we start training at the first $\epsilon$ value that presented CO for $S - 1$ steps. We report the average final PGD-20 test accuracy over 3 random seeds for every $\epsilon$ value.

In Fig. 5, we show that the phase transition leading to CO is observed in MNIST, SVHN and CIFAR10 for progressively larger $\epsilon_c$ for larger $S$. The phase transition is depicted by the abrupt decay of the PGD-20

Acc. to zero in a small $\epsilon$ change. This aligns with our insights from Thms 4.1 and 4.3 and with previous experimental evaluations (Andriushchenko & Flammarion, 2020).

Our analysis in the toy model shows the existence of a CO solution with lower loss than the robust solution for larger $\epsilon$. Recent works argue longer training schedules might lead to CO (Kim et al., 2021; Abad Rocamora et al., 2024). In relation to our analysis, longer schedules might converge to the CO solutions shorter schedules did not. In Sec. C.4 we find that $\epsilon_c$ can be smaller in certain datasets for longer schedules.

### 6.3 The effect of weight decay

In Secs. 4 and 5 we demonstrated that CO solutions exist for any dataset, $S$ and $\epsilon$. Provided these solutions exist, if the complexity of our model is not controlled, we might converge to fitting one of the CO solutions. A popular strategy for constraining the complexity of a model is weight decay (Hanson & Pratt, 1988; Krogh & Hertz, 1991).

In this section, we study the effectiveness of weight decay for avoiding CO. We perform a grid-search over different $\lambda_{\mathrm{wd}}$ values, searching for the ones which do not present CO (see Sec. C.2). We train PreActResNet18 on CIFAR10 with the standard $\lambda_{\mathrm{wd}} = 0.0005$ and $\lambda_{\mathrm{wd}} = 0.01$. We employ the local linearity regularization term ELLE-A (Abad Rocamora et al., 2024) as a reference. For SVHN experiments, we refer to Sec. C.2.

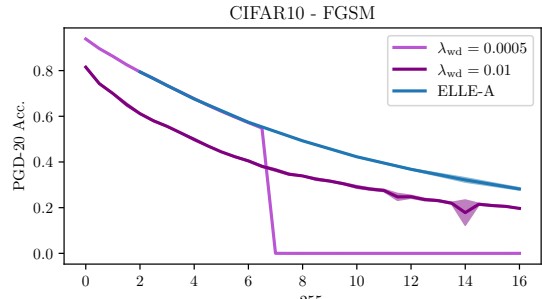

In Fig. 6, we observe that with $\lambda_{\mathrm{wd}} = 0.005$ CO appears at $\epsilon_c = 6.5/255$. Contrarily, for $\lambda_{\mathrm{wd}} = 0.01$, we avoid CO and obtain a non-zero PGD-20 accuracy for all $\epsilon$. However, this performance is significantly lower than when employing the ELLE-A regularizer. We conclude that constraining the complexity of the classifier through regularization can indeed avoid CO, aligning with our theoretical insights in the toy model (Thm 4.3). However, while weight decay avoids CO, the performance can be more than 10 accuracy points worse than regularization terms tailored for AT like ELLE-A.

Figure 6: **Weight decay and local linearity regularization:** We train PreActResNet18 on CIFAR10 with $\lambda_{\mathrm{wd}} \in \{0.0005, 0.01\}$ and the ELLE-A local linearity regularization. Constraining the model complexity with regularization helps avoiding CO, with specialized regularizers like ELLE-A performing best.

## 7 Conclusions

To the best of our knowledge, this work is the first to relate CO to a phase transition in the adversarial budget $\epsilon$. Beyond a critical threshold $\epsilon_c$, non-robust solutions become optimal, even for multi-step PGD attacks or randomized initialization. Our findings, obtained through theoretical models and verified by large-scale experiments, challenge the notion that CO can be fully avoided by simply increasing the number of PGD steps or adding noise during training.

Furthermore, we demonstrated that regularization techniques, such as weight decay, can effectively mitigate CO by constraining the solution space, albeit with potential trade-offs in model performance, explaining the success of strategies such as local linearity regularization.

**Limitations and future work:** While our main theorems establish the existence of solutions exhibiting CO, the practical manifestation of CO is also shaped by training dynamics. For instance, as observed in Sec. C.4, variations in training duration can influence the critical adversarial budget $\epsilon_c$. We did not incorporate training dynamics into our analysis.

In Sec. 3, we find CO arises abruptly for $\epsilon > \epsilon_c$ as a consequence of a phase transition. While the phase transition is fully characterized in two different models (Secs. A.2 and 3.1), the phase transition in larger scales can only be demonstrated empirically, e.g., Fig. 5. Characterizing the effect of optimizers, datasets and NN architectures in CO is still an open problem.

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

Table 1: **Asset licenses**

| Asset | License |
|---|---|
| MNIST (LeCun et al., 1998) | Creative Commons Attribution-Share Alike 3.0 |
| CIFAR10 (Krizhevsky, 2009) | MIT |
| SVHN (Netzer et al., 2011) | CC0:Public Domain |
| torch (Paszke et al., 2017) | BSD 3-Clause "New" or "Revised" License |
| timm (Wightman, 2019) | Apache License 2.0 |

## Contents of the appendix

In Table 1 we cover the licenses of the assets employed in this work. In Sec. A we provide additional derivations for our toy model. Next, in Sec. B we present our proofs and finally, in Sec. C, we present additional experimental results.

## A   Additional details on toy models for Catastrophic Overfitting

In Sec. A.1 we provide additional details regarding the sinusoidal toy model studied in the text. In Sec. A.2, we present an additional toy model presenting the same behavior as described in the main text and repeat the experiments in Sec. 6.1 with this additional model.

### A.1   Sinusoidal toy model

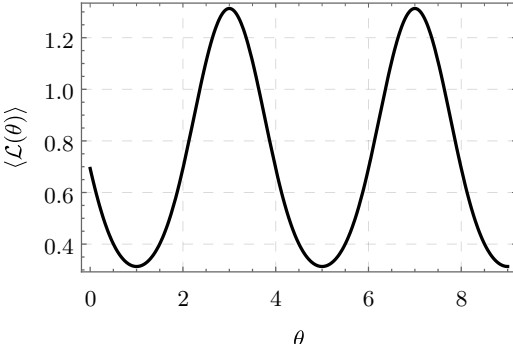 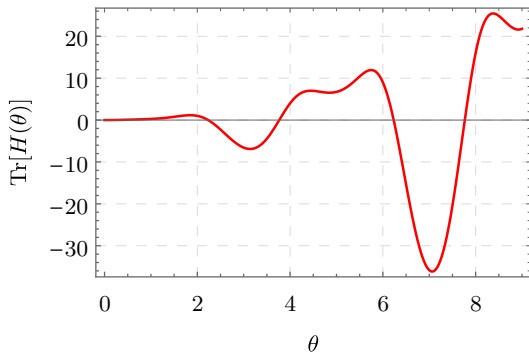

Figure 7: **Loss landscape and sample Hessian for the toy model and $\epsilon = 0$:** On the *left*, we plot the loss function with respect to the single parameter in the model $\theta$. On the *right*, we plot the Hessian with respect to the samples. As described in the main text, the loss has infinite degenerate extrema, where the curvature term, proportional to sample Hessian, grows as $\theta^2$.

Here, we provide a detailed analysis of the toy model presented in the main text, under a single step adversarial attack.

We consider a training dataset consisting of two points $x_1 = -x_2 \in \mathbb{R}$, where we choose $x_{1,2} = \pm\pi/2$. The task is binary classification, with labels generated by taking $y = \Theta(x)$, where $\Theta(z)$ is the Heaviside function. We use the cross-entropy (CE) loss $\langle \mathcal{L} \rangle = \frac{1}{2} \sum_{i=1}^{2} \mathcal{L}(f_\theta(x_i), y_i)$, where $\mathcal{L}(f_\theta(x_i), y_i) = -y_i \log p_i - (1 - y_i) \log(1 - p_i)$, and the probabilities are simply $p_i = e^{f_\theta(x_i)}/(1 + e^{f_\theta(x_i)})$. We take the network function to be

$$f_\theta(x_i) = \sin(\theta x_i), \qquad \theta \in \mathbb{R}^+. \tag{10}$$

where $\theta$ is the single network parameter.

The loss is given by the mean over the single sample losses taken on the two pairs of samples and labels $(x_1, y_1) = (x_1, 1)$ and $(x_2, y_2) = (x_2, 0)$, as

$$\langle \mathcal{L} \rangle = \frac{1}{2} \sum_{i=1}^{2} \mathcal{L}(f_\theta(x_i), y_i) = \frac{1}{2} \left( -\log \left( \frac{1}{1 + e^{-\sin(\theta x_1)}} \right) - \log \left( \frac{1}{1 + e^{\sin(\theta x_2)}} \right) \right) \tag{11}$$

$$= -\log \left( \frac{1}{1 + e^{-\sin\left(\frac{\pi\theta}{2}\right)}} \right).$$

This loss is minimized when $\partial_\theta \langle \mathcal{L} \rangle = 0$, which gives $\cos\left(\frac{\pi\theta}{2}\right) = 0$, satisfied for the set of degenerate minima $\theta_{\min} = 4n + 1$, and a set of maxima at $\theta_{\max} = 4n + 3$ where $n \in \mathbb{Z}^+$. Under an FGSM attack with parameter $\epsilon$, the effective loss being optimized is $\mathcal{L}(f_\theta(x_i + \delta_i), y_i)$, where

$$\delta_t^i = \epsilon \cdot \text{sign}\left( \nabla_{x_i} \mathcal{L}(f_{\theta_t}(x_i), y_i) \right) = \epsilon \, \text{sign} \begin{pmatrix} -\dfrac{\theta \cos(\theta x_1)}{2e^{\sin(\theta x_1)} + 2} \\ \dfrac{\theta \cos(\theta x_2)}{2e^{-\sin(\theta x_2)} + 2} \end{pmatrix}, \tag{12}$$

and is given explicitly by

$$\langle \mathcal{L} \rangle = \frac{1}{2} \sum_{i=1}^{2} \mathcal{L}(f_\theta(x_i + \delta_i), y_i) = \log \left( 1 + e^{-\sin\left(\theta\left(\frac{\pi}{2} - \epsilon \, \text{sign}\left(\theta \cos\left(\frac{\theta\pi}{2}\right)\right)\right)\right)} \right). \tag{13}$$

The extrema landscape of the pertrubed loss differs from the original loss, increasingly so as $\epsilon$ is increased. While the perturbed loss can be fully described numerically, in order to gain analytical insights, it is worthwhile to Taylor expand the loss function for $\epsilon \ll 1$ as

$$\langle \mathcal{L}(\theta, \epsilon) \rangle = \frac{1}{2} \sum_{i=1}^{2} \sum_{n=0}^{\infty} \epsilon^n \frac{\partial_{x_i}^n \mathcal{L}(\theta)}{n!} = \log \left( 1 + e^{-\sin\left(\frac{\pi\theta}{2}\right)} \right) \tag{14}$$

$$+ \epsilon \left| \frac{\theta \cos\left(\frac{\pi\theta}{2}\right)}{1 + e^{\sin\left(\frac{\pi\theta}{2}\right)}} \right| + \epsilon^2 \, \text{sign} \left( \theta \cos\left(\frac{\pi\theta}{2}\right) \right)^2 H(\theta) + \mathcal{O}(\epsilon^3),$$

which amounts to the result given in Eq. (4) presented in the the main text when plugging in $x_{1,2} = \pm\frac{\pi}{2}$, where we defined the sample curvature as

$$H(\theta) = \partial_{x_i}^2 \mathcal{L}(f_\theta(x_i), y_i) = \begin{cases} \dfrac{\theta^2 \left( \sin(\theta x_1) + e^{\sin(\theta x_1)} (\sin(\theta x_1) + \cos^2(\theta x_1)) \right)}{2 \left( 1 + e^{\sin(\theta x_1)} \right)^2} \\ -\dfrac{\theta^2 e^{\sin(\theta x_2)} \left( (1 + e^{\sin(\theta x_2)}) \sin(\theta x_2) - \cos^2(\theta x_2) \right)}{2 \left( 1 + e^{\sin(\theta x_2)} \right)^2} \end{cases}. \tag{15}$$

## A.2 ReLU toy model

In this section, we propose a model with ReLU activations where the phase transition resulting in CO can also be observed. Let the training points, loss function and $\theta \in [0, \theta_{\max}]$ be defined as in Sec. 3.1, we define the classifier to be:

$$f_\theta(x) = x + \theta \cdot g_{\delta, \beta}(x), \tag{16}$$

where:

$$g_{\delta, \beta}(x) = -x - \beta \cdot (-\text{ReLU}(x - (\pi/2 + \delta)) + \text{ReLU}(-x - (\pi/2 + \delta))), \tag{17}$$

with real numbers $\beta \geq 0$ and $\delta > 0$. In practice, we fix $\beta = 5$ and $\delta = 10^{-10}$. Intuitively, our classifier $f$ can be thought of being a composition of a robust classifier $(x)$ and a non-robust classifier $(g_{\delta, \beta}(x))$. These two classifiers and their losses with respect to $x$ are visualized in Fig. 8.

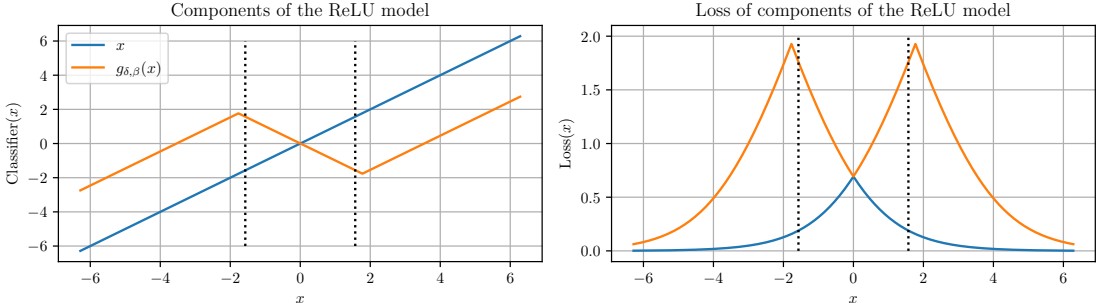

Figure 8: **Visualization of the components of Eq. (16) and their loss with $\beta = 2$ and $\delta = 0.2$:** The dotted lines highlight the points $\pm\pi/2$. The larger $\theta$ is, the more weight is put on the non-robust classifier $g_{\delta,\beta}(x)$, getting away from the robust solution $f_\theta(x) = x$.

We can observe that the function $g_{\delta,\beta}(x)$ presents positive value at $x = -\pi/2$ and negative value at $x = \pi/2$, being a bad classifier for the clean samples. However, for smaller $x$ values, the classifier becomes negative and for larger $x$ values the function becomes positive. Since $\frac{d}{dx}\mathcal{L}(g_{\delta,\beta}(x), y)\big|_{x=-\pi/2, y=0} < 0$ and $\frac{d}{dx}\mathcal{L}(g_{\delta,\beta}(x), y)\big|_{x=\pi/2, y=1} > 0$, the FGSM attack will lead us to points with smaller loss than the original ones, this will enable the appearance of CO.

Now, we repeat the experiment done in Sec. 6.1, where we compute the optimal classifier for FGSM AT and the true AT loss (Eq. (AT)), for the classifier in Eq. (16).

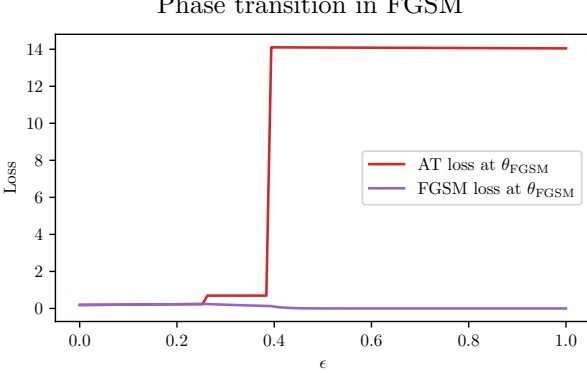

Figure 9: **Phase transition in the ReLU toy model:** The critical values are $\epsilon_{c1} = \epsilon \approx 0.25$ and $\epsilon_{c2} = \epsilon \approx 0.39$. We evaluate the FGSM and AT losses at the optimal weight for the FGSM loss, i.e., $\theta_{\text{FGSM}}$. For $\epsilon > \epsilon_{ci}$, the phase transition occurs, resulting in a sudden increase in the AT loss and a decrease in the FGSM loss.

In Fig. 9, we can observe that two phase transitions occur, one at $\epsilon \approx 0.25$ and the other at $\epsilon \approx 0.39$, leading to the AT and FGSM losses to be separate further with every phase transition.

In Fig. 10, we can observe that for $\epsilon = 0.1$ both the FGSM and AT solutions coincide at $\theta = 0$, for $\epsilon = 0.3$, after the first phase transition, the optimal solution for FGSM is $\theta_{\text{FGSM}} = 1$. Finally, for $\epsilon = 1$, after the second phase transition, the optimal classifier for FGSM becomes $\theta_{\text{FGSM}} = \theta_{\max} = 10$.

## B   Proofs

*Proof of Thm 4.1.* Our proof flows as:

    i) We show that the optimal loss value is attained when $\sin(\theta \cdot (x_i + \delta_S^i)) = y_i$.

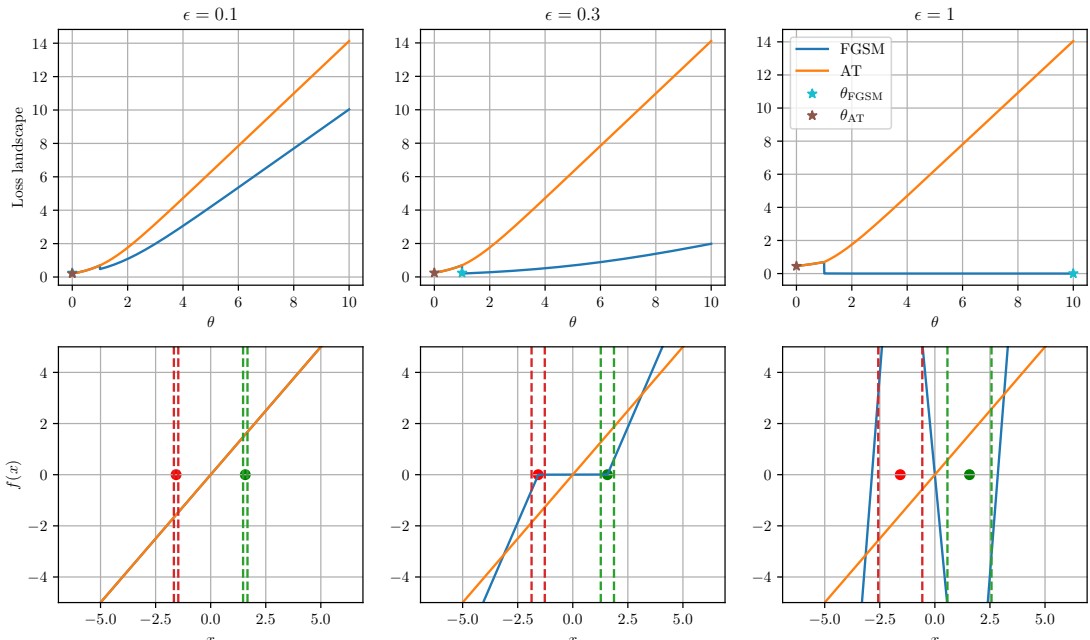

Figure 10: **Loss landscape and optimal classifiers for the ReLU toy model and** $\epsilon \in \{0.1, 0.3, 1\}$**:** In the *top row*, we plot the loss function with respect to the single parameter in the model $\theta$. In the *bottom row*, we plot the optimal classifiers according to the loss in the upper row, the two training points (🔴 and 🟢) and their adversarial regions (-- and --). The optimal classifier obtained with FGSM (⭐) coincides with one obtained with AT (⭐) for $\epsilon = 0.1$. For $\epsilon > \epsilon_{c1}$, the FGSM and AT solutions differ, with the optimal solution for FGSM being $\theta_{\mathrm{FGSM}} = 1$. Similarly, for $\epsilon > \epsilon_{c2}$, FGSM becomes $\theta_{\mathrm{FGSM}} = 10$.

ii) We look for $\theta_k$ so that:

$$\sin(\theta_k \cdot (x_i - \rho)) = \sin\left(\theta_k \cdot \left(x_i - \rho + \frac{\epsilon_k}{S}\right)\right) = y_i \,,$$

and form $\rho, \epsilon_k$ and $\theta_k$ depending on $S, a, b_k, k$.

iii) We show that $|\mathcal{L}(\sin(\theta_k \cdot (x_i - \rho)), y_i) - \mathcal{L}(\sin(\theta_k \cdot x_i), y_i)| \leq 2 \cdot \theta_k \cdot \rho$.

Because of the symmetry of the sin function and our training points $\{(-\pi/2, -1), (\pi/2, 1)\}$, we will continue the analysis just by looking at the loss at $(\pi/2, 1)$.

Starting with $i$), let $\mathcal{L}(f_{\theta_k}(\pi/2), 1) = -\sin(\theta_k \cdot (\pi/2 + \delta_S)) + \log\left(1 + e^{\sin(\theta_k \cdot (\pi/2 + \delta_S))}\right)$. It is easy to see that the optimal loss value is $\mathcal{L}^\star = \log(1 + e) - 1 \approx 0.3133$, by minimizing $\mathcal{L}(f_{\theta_k}(\pi/2), 1)$ as a function of $\delta_S$, where $\delta_S = \frac{\pi(2n+1) - \frac{\pi}{2}}{\theta} - \frac{\pi}{2}$.

Following with $ii$), our goal is to obtain $\theta_k$ so that $\sin(\theta_k \cdot (x - \rho + \delta_S)) = 1$. In the case $\rho = 0$ it is trivial that, since $\frac{d}{dx}\sin(\theta_k \cdot (\pi/2)) = \theta_k \cdot \cos(\theta_k \cdot \pi/2)$, by setting $\theta_k = 1$, we obtain $\delta_1 = \mathrm{sign}(\cos(\pi/2)) = \mathrm{sign}(0) = 0$ and therefore $\delta_S = 0$. We study $\rho > 0$ as it is a more realistic scenario where the adversary "moves".

We are then set with the problem of finding:

$$\sin(\theta_k \cdot (\pi/2 - \rho + \delta_S)) = 1 \,.$$

To ease the analysis, we will further impose that:

$$\sin(\theta_k \cdot (\pi/2 - \rho + \delta_s)) = 1 \,, \forall s = 0, \cdots, S \,.$$

Then it is enough to look for:

$$\sin(\theta_k \cdot (\pi/2 - \rho)) = \sin\left(\theta_k \cdot \left(\pi/2 - \rho + \frac{\epsilon_k}{S}\right)\right) = 1 \,, \tag{18}$$

as the PGD perturbations at each step will just be $\delta_s = \frac{\epsilon_k \cdot s}{S}$. In order to satisfy Eq. (18), the minimal $\theta_k$ must be the one where the two maxima of the sin at $\theta_k \cdot (\pi/2 - \rho)$ and $\theta_k \cdot \left(\pi/2 - \rho + \frac{\epsilon_k}{S}\right)$ are contiguous, i.e., there is no other maximum in between them. For this to happen, we just need to find:

$$\left. \begin{array}{rcl} \theta_k \cdot (\pi/2 - \rho) & = & \pi/2 + 2 \cdot \pi \cdot k \\ \theta_k \cdot \frac{\epsilon_k}{S} & = & 2 \cdot \pi \end{array} \right\} , \tag{19}$$

For some $k = 0, \cdots, \infty$. Then, by setting $\epsilon_k = \frac{2 \cdot \pi \cdot S}{b_k}$ and $\rho = \frac{\pi}{2 \cdot a}$, we have from the second equation $\theta_k = b_k$. We can substitute into the first equation of Eq. (19) and solve for $b_k$:

$$b_k = \frac{1 + 4 \cdot k}{1 - \frac{1}{a}} .$$

It is then easy to see that

$$\sin(\theta_k \cdot (\pi/2 - \rho + \epsilon_k)) = \sin\left( \frac{1 + 4 \cdot k}{1 - \frac{1}{a}} \cdot (\pi/2 - \frac{\pi}{2 \cdot a} + \frac{(2 \cdot \pi \cdot S) \cdot (1 - \frac{1}{a})}{1 + 4 \cdot k}) \right) \tag{20}$$

$$= \sin\left( \pi/2 \cdot \frac{1 + 4 \cdot k}{1 - \frac{1}{a}} \cdot (1 - \frac{1}{a} + \frac{(4 \cdot S) \cdot (1 - \frac{1}{a})}{1 + 4 \cdot k}) \right)$$

$$= \sin\left( \pi/2 \cdot (1 + 4 \cdot k + (4 \cdot S)) \right) = 1.$$

And that similarly $\sin(\theta_k \cdot (\pi/2 - \rho + \frac{\epsilon_k}{2 \cdot S})) = -1$, which constitutes an adversarial example.

Finally, we prove item iii):

$$\left| \mathcal{L}\left( \sin(\theta_k \cdot (\pi/2 - \rho)), 1 \right) - \mathcal{L}\left( \sin(\theta_k \cdot \pi/2), 1 \right) \right| \le \rho \cdot \max_{x \in \mathbb{R}} \frac{d}{dx} \mathcal{L}\left( \sin(\theta_k \cdot x), 1 \right)$$

$$= \rho \cdot \max_{x \in \mathbb{R}} \theta_k \cdot \left| \cos(\theta_k \cdot x) \left( \frac{e^{\sin(\theta_k \cdot x)}}{1 + e^{\sin(\theta_k \cdot x)}} - 1 \right) \right|$$

$$[|\cos(\theta_k \cdot x)| \le 1 \text{ and } \left| \frac{e^{\sin(\theta_k \cdot x)}}{1 + e^{\sin(\theta_k \cdot x)}} - 1 \right| \le 2] \le 2 \cdot \rho \cdot \theta_k$$

$$= \frac{\pi \cdot b_k}{a} ,$$

where in the first line, we used the Taylor remainder of the perturbed loss. This concludes the proof. $\qquad\square$

*Proof of Thm C.1.* Let the AT problem be as in Eq. (AT):

$$\min_{\boldsymbol{\theta}_\alpha} \frac{1}{n} \sum_{i=1}^{n} \max_{||\boldsymbol{\delta}_i||_\infty \le \epsilon_\alpha} \mathcal{L}(\boldsymbol{f}_{\boldsymbol{\theta}}(\alpha \cdot \boldsymbol{x}_i + \boldsymbol{\delta}_i), y_i) = \min_{\hat{\boldsymbol{\theta}}_\alpha, \boldsymbol{W}_\alpha, \boldsymbol{b}_\alpha} \frac{1}{n} \sum_{i=1}^{n} \max_{||\boldsymbol{\delta}_i||_\infty \le \epsilon_\alpha} \mathcal{L}(\hat{\boldsymbol{f}}_{\hat{\boldsymbol{\theta}}_\alpha}(\boldsymbol{W}_\alpha(\alpha \cdot \boldsymbol{x}_i + \boldsymbol{\delta}_i) + \boldsymbol{b}_\alpha), y_i)$$

$$= \min_{\hat{\boldsymbol{\theta}}_\alpha, \boldsymbol{W}_\alpha, \boldsymbol{b}_\alpha} \frac{1}{n} \sum_{i=1}^{n} \max_{||\boldsymbol{\delta}_i||_\infty \le \epsilon_\alpha} \mathcal{L}(\hat{\boldsymbol{f}}_{\hat{\boldsymbol{\theta}}_\alpha}(\alpha \cdot \boldsymbol{W}_\alpha(\boldsymbol{x}_i + \frac{1}{\alpha}\boldsymbol{\delta}_i) + \boldsymbol{b}_\alpha), y_i)$$

$$= \min_{\hat{\boldsymbol{\theta}}_\alpha, \boldsymbol{W}_\alpha, \boldsymbol{b}_\alpha} \frac{1}{n} \sum_{i=1}^{n} \max_{||\boldsymbol{\delta}_i||_\infty \le \epsilon_\alpha/\alpha} \mathcal{L}(\hat{\boldsymbol{f}}_{\hat{\boldsymbol{\theta}}_\alpha}(\alpha \cdot \boldsymbol{W}_\alpha(\boldsymbol{x}_i + \boldsymbol{\delta}_i) + \boldsymbol{b}_\alpha), y_i)$$

$$[\boldsymbol{W} = \alpha \cdot \boldsymbol{W}_\alpha, \boldsymbol{b} = \boldsymbol{b}_\alpha, \boldsymbol{\theta} = \boldsymbol{\theta}_\alpha \text{ and } \epsilon = \epsilon/\alpha] = \min_{\hat{\boldsymbol{\theta}}, \boldsymbol{W}, \boldsymbol{b}} \frac{1}{n} \sum_{i=1}^{n} \max_{||\boldsymbol{\delta}_i||_\infty \le \epsilon} \mathcal{L}(\hat{\boldsymbol{f}}_{\hat{\boldsymbol{\theta}}}(\boldsymbol{W}(\boldsymbol{x}_i + \boldsymbol{\delta}_i) + \boldsymbol{b}), y_i)$$

$$= \min_{\boldsymbol{\theta}} \frac{1}{n} \sum_{i=1}^{n} \max_{||\boldsymbol{\delta}_i||_\infty \le \epsilon} \mathcal{L}(\boldsymbol{f}_{\boldsymbol{\theta}}(\boldsymbol{x}_i + \boldsymbol{\delta}_i), y_i) .$$

$$\tag{21}$$

This shows that performing AT in the dataset $\{(\boldsymbol{x}_i \cdot \alpha, y_i)\}_{i=1}^{n}$ with $\epsilon_\alpha$ is effectively the same as performing AT on the standard dataset $\{(\boldsymbol{x}_i, y_i)\}_{i=1}^{n}$ with $\epsilon = \epsilon_\alpha/\alpha$. $\qquad\square$

*Proof of Thm C.2.* As proved in Thm C.1, performing AT in the dataset $\{(\boldsymbol{x}_i \cdot \alpha, y_i)\}_{i=1}^n$ with $\epsilon_\alpha$ is effectively the same as performing AT on the standard dataset $\{(\boldsymbol{x}_i, y_i)\}_{i=1}^n$ with $\epsilon = \epsilon_\alpha/\alpha$.

This means that if in the standard dataset $\{(\boldsymbol{x}_i, y_i)\}_{i=1}^n$ we have $(\beta, \eta)$-CO for $\epsilon > \epsilon_c$, since rescaling the dataset results in an effective adversarial budget of $\epsilon_\alpha = \epsilon \cdot \alpha$, in the re-scaled dataset $\{(\boldsymbol{x}_i \cdot \alpha, y_i)\}_{i=1}^n$ we will have $(\beta, \eta)$-CO for $\epsilon > \epsilon_c \cdot \alpha$. $\qquad\square$

*Proof of Thm 4.2.* Based on Thm 2.1, we need to show:

  i) Nearly perfect accuracy in the PGD perturbations $\delta_S^i$.

  ii) Close-to-zero accuracy in any other points with $|\delta_\star^i| \le \epsilon_k$.

Since we are going to make arguments when $a \to \infty$, we will make the dependence of every variable with respect to $a$ explicit, e.g., $\theta_k(a)$. Our result will hold with $\beta = \eta = 0$ in Thm 2.1. The condition i) is obtained by construction of Thm 4.1, where we showed that:

$$\sin\left(\theta_k(a) \cdot \left(x_i - \frac{\pi}{2 \cdot a} + \epsilon_k(a)\right)\right) = y_i \quad \forall i \in \{1, 2\}, \tag{22}$$

which means that because $\frac{d\sin(\theta \cdot x)}{dx} = \theta \cdot \cos(\theta \cdot x) \le \theta$, by Lipchitzness:

$$\left| \sin\left(\theta_k(a) \cdot \left(x_i - \frac{\pi}{2 \cdot a} + \epsilon_k(a)\right)\right) - \sin\left(\theta_k(a) \cdot (x_i + \epsilon_k(a))\right) \right| \le \theta_k(a) \cdot \frac{\pi}{2 \cdot a}$$

$$= \frac{1 + 4 \cdot k}{1 - \frac{1}{a}} \cdot \frac{\pi}{2 \cdot a} \quad \forall i \in \{1, 2\},$$

which implies $\lim_{a \to \infty} \sin\left(\theta_k(a) \cdot (x_i + \epsilon_k(a))\right) = y_i \quad \forall i \in \{1, 2\}$. Similarly, we can show ii). Firstly, let us see that:

$$\sin\left(\theta_k(a) \cdot \left(x_i - \frac{\pi}{2 \cdot a} \pm \frac{\epsilon_k(a)}{2 \cdot S}\right)\right) = -y_i \quad \forall i \in \{1, 2\}.$$

Starting from the definition and using the fact that $\epsilon_k(a) = \frac{2\pi S}{\theta_k(a)}$ and Eq. (22):

$$\sin\left(\theta_k(a) \cdot \left(x_i - \frac{\pi}{2 \cdot a} + \frac{\epsilon_k(a)}{2 \cdot S}\right)\right) = \sin\left(\theta_k(a) \cdot \left(x_i - \frac{\pi}{2 \cdot a} + \epsilon_k(a) \pm \frac{(2 \cdot S - 1) \cdot \epsilon_k(a)}{2 \cdot S}\right)\right)$$

$$\left[\epsilon_k(a) = \frac{2\pi S}{\theta_k(a)}\right] = \sin\left(\theta_k(a) \cdot \left(x_i - \frac{\pi}{2 \cdot a} + \epsilon_k(a)\right) \pm (2 \cdot S - 1) \cdot \pi\right)$$

$$\left[\sin(x \pm p \cdot \pi) = -\sin(x) \text{ for odd } p\right] = -\sin\left(\theta_k(a) \cdot \left(x_i - \frac{\pi}{2 \cdot a} + \epsilon_k(a)\right)\right)$$

$$\left[\text{Eq. (22)}\right] = -y_i \quad \forall i \in \{1, 2\}.$$

Then, using the same arguments as before:

$$\lim_{a \to \infty} \sin\left(\theta_k(a) \cdot \left(x_i \pm \frac{\epsilon_k(a)}{2 \cdot S}\right)\right) = -y_i \quad \forall i \in \{1, 2\},$$

which shows that the points $x_i \pm \frac{\epsilon_k(a)}{2 \cdot S}$ are classified with the wrong label. By showing i) and ii), we have shown that $(0, 0)$-CO occurs with arbitrarily small $\epsilon_k(a)$, large $S$ and with arbitrarily accurate solutions when increasing $a$. $\qquad\square$

*Proof of Thm 4.3.* It is easy to check that if $|b_k| \le B$ and $b_k \ge 0$, we have that:

$$\epsilon_k := \frac{2\pi S}{b_k} \ge \frac{2\pi S}{B}.$$

Then, by increasing $S$, we can increase the lower bound on $\epsilon_k$ and therefore, there will not be solutions in the form of Thm 4.1. $\qquad\square$

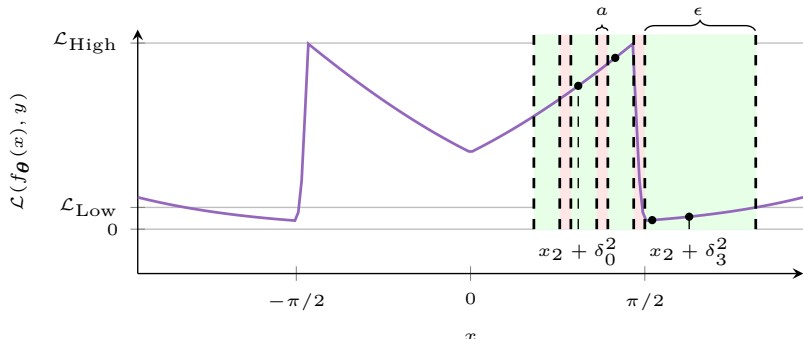

Figure 11: **Example and proof sketch for Thms 5.1 and 5.3:** Example of the loss for $S = 3$, $\epsilon = 1$, $a = 0.1$, $b = 4$ and $\delta_0^2 = -0.6$. If $\delta_0^2$ falls in a green region ($\mathcal{D}_{\text{Low}}$), the loss after the PGD attack can be upper bounded by $\mathcal{L}_{\text{Low}}$. If $\delta_0^2$ falls in a red region ($\mathcal{D}_{\text{High}}$), the loss after the PGD attack can be upper bounded by $\mathcal{L}_{\text{High}}$. By making $a$ smaller, we reduce the probability of falling in a red zone. Then, by increasing $b$, we reduce $\mathcal{L}_{\text{Low}}$ and we reduce the overall expected loss.

*Proof of Thm 5.1.* Our goal is to upper bound:

$$\frac{1}{2} \sum_{i=1}^{2} \mathbb{E}_{\delta_0^i \sim \text{Unif.}([-\epsilon,\epsilon])} \left[ \mathcal{L}(f_{\boldsymbol{\theta}}(x_i + \delta_S^i), y_i) \right] = \mathbb{E}_{\delta_0^2 \sim \text{Unif.}([-\epsilon,\epsilon])} \left[ \mathcal{L}(f_{\boldsymbol{\theta}}(x_2 + \delta_S^2), y_2) \right] , \tag{23}$$

because of symmetry in the data and the classifier. In order to upper bound this expectation, we decouple the loss into two terms: (i) The expected loss when $\delta_S^2 \in [0, \epsilon]$. (ii) The expected loss when $\delta_S^2 \notin [0, \epsilon]$.

It is easy to see that the only way for $\delta_S^2 \notin [0, \epsilon]$ to happen is if $\delta_s^2 \in [-a, 0]$ for some $s \in \{0, 1, \cdots, S\}$. If this happens at some point of the trajectory, the PGD trajectory will get stuck in $[-\epsilon/S - a, -\epsilon/S] \cup [-a, 0]$. Since $\frac{d}{d\delta} \mathcal{L}(f_{\boldsymbol{\theta}}(x_2 + \delta), y_2) \geq 0 \;\; \forall \delta \in [0, \epsilon]$, the only way to fall in $[-a, 0]$ at some point of the trajectory is to arrive from $\delta_0^2 \in [-\epsilon, 0]$. Moreover, since the stepsizes are $1/S$, we have that for $\delta_S^2 \notin [0, \epsilon]$ to happen, we need $\delta_0^2 \in [-\frac{\epsilon \cdot (S-1)}{S} - a, -\frac{\epsilon \cdot (S-1)}{S}] \cup \cdots \cup [-\frac{\epsilon}{S} - a, -\frac{\epsilon}{S}] \cup [-a, 0] = \mathcal{D}_{\text{High}}$. For a visual sketch please check Fig. 11.

Then, Eq. (23) becomes:

$$\frac{1}{2} \sum_{i=1}^{2} \mathbb{E}_{\delta_0^i \sim \text{Unif.}([-\epsilon,\epsilon])} \left[ \mathcal{L}(f_{\boldsymbol{\theta}}(x_i + \delta_S^i), y_i) \right] = P\left(\delta_0^2 \in \mathcal{D}_{\text{Low}}\right) \cdot \mathbb{E}_{\delta_0^2 \sim \text{Unif.}([-\epsilon,\epsilon])} \left[ \mathcal{L}(f_{\boldsymbol{\theta}}(x_2 + \delta_S^2), y_2)\big|_{\delta_0^2 \in \mathcal{D}_{\text{Low}}} \right]$$

$$+ P\left(\delta_0^2 \in \mathcal{D}_{\text{High}}\right) \cdot \mathbb{E}_{\delta_0^2 \sim \text{Unif.}([-\epsilon,\epsilon])} \left[ \mathcal{L}(f_{\boldsymbol{\theta}}(x_2 + \delta_S^2), y_2)\big|_{\delta_0^2 \in \mathcal{D}_{\text{High}}} \right]$$

$$\leq P\left(\delta_0^2 \in \mathcal{D}_{\text{Low}}\right) \cdot \max_{\delta_0^2 \in \mathcal{D}_{\text{Low}}} \mathcal{L}(f_{\boldsymbol{\theta}}(x_2 + \delta_S^2), y_2)$$

$$+ P\left(\delta_0^2 \in \mathcal{D}_{\text{High}}\right) \cdot \max_{\delta_0^2 \in \mathcal{D}_{\text{High}}} \mathcal{L}(f_{\boldsymbol{\theta}}(x_2 + \delta_S^2), y_2) .$$

$$= P\left(\delta_0^2 \in \mathcal{D}_{\text{Low}}\right) \cdot \mathcal{L}(f_{\boldsymbol{\theta}}(x_2 + \epsilon), y_2) .$$

$$+ \left(1 - P\left(\delta_0^2 \in \mathcal{D}_{\text{Low}}\right)\right) \cdot \mathcal{L}(f_{\boldsymbol{\theta}}(x_2 - \alpha), y_2) .$$

Then, it is only left to compute $P\left(\delta_0^2 \in \mathcal{D}_{\text{Low}}\right)$, $\mathcal{L}(f_{\boldsymbol{\theta}}(x_2 + \epsilon), y_2)$ and $\mathcal{L}(f_{\boldsymbol{\theta}}(x_2 - \alpha), y_2)$. It is easy to check that for $\alpha = 1/S$, we have $P\left(\delta_0^2 \in \mathcal{D}_{\text{Low}}\right) = \frac{2 \cdot \epsilon - S \cdot a}{2 \cdot \epsilon}$ and for for $\alpha = 2/S$, we have $P\left(\delta_0^2 \in \mathcal{D}_{\text{Low}}\right) = \frac{2 \cdot \epsilon - \lceil \frac{S}{2} \rceil \cdot a}{2 \cdot \epsilon} \geq$

$\frac{2 \cdot \epsilon - S \cdot a}{2 \cdot \epsilon}$. Next, for computing the losses, we compute the value of the classifier:

$$
\begin{aligned}
f_{\boldsymbol{\theta}}(x_2 + \epsilon) &= h(x_2 + \epsilon; x_2, a, b, \pi/2) \\
&= -\mathrm{ReLU}(\pi/2 + \epsilon + \pi/2 - \pi/2) \\
&\quad + (1 + b/a) \cdot \mathrm{ReLU}(\pi/2 + \epsilon - \pi/2 + a) \\
&\quad - (1 + b/a) \cdot \mathrm{ReLU}(\pi/2 + \epsilon - \pi/2) \\
&= -(\pi/2 + \epsilon) + (1 + b/a) \cdot (\epsilon + a) - (1 + b/a) \cdot \epsilon \\
&= -\pi/2 + a + b - \epsilon \,.
\end{aligned}
\tag{24}
$$

Similarly, for $x_2 - a$:

$$
\begin{aligned}
f_{\boldsymbol{\theta}}(x_2 - a) &= h(x_2 - a; x_2, a, b, \pi/2) \\
&= -\mathrm{ReLU}(\pi/2 - a + \pi/2 - \pi/2) \\
&\quad + (1 + b/a) \cdot \mathrm{ReLU}(\pi/2 - a - \pi/2 + a) \\
&\quad - (1 + b/a) \cdot \mathrm{ReLU}(\pi/2 - a - \pi/2) \\
&= -(\pi/2 - a) + (1 + b/a) \cdot 0 + (1 + b/a) \cdot 0 \\
&= -\pi/2 + a \,.
\end{aligned}
\tag{25}
$$

Then, the loss becomes $\mathcal{L}(f_{\boldsymbol{\theta}}(x_2 + \epsilon), y_2) = \log\left(1 + e^{\pi/2 + \epsilon - a - b}\right) \leq \log\left(1 + e^{\pi/2 + \epsilon - b}\right)$ and $\mathcal{L}(f_{\boldsymbol{\theta}}(x_2 - a), y_2) = \log\left(1 + e^{\pi/2 - a}\right) \leq \log\left(1 + e^{\pi/2}\right)$ and finally, we obtain the desired result:

$$
\begin{aligned}
\frac{1}{2} \sum_{i=1}^{2} \mathop{\mathbb{E}}_{\delta_0^i \sim \mathrm{Unif.}([-\epsilon, \epsilon])} \left[ \mathcal{L}(f_{\boldsymbol{\theta}}(x_i + \delta_S^i), y_i) \right] &\leq \frac{2 \cdot \epsilon - S \cdot a}{2 \cdot \epsilon} \cdot \log\left(1 + e^{\pi/2 + \epsilon - b}\right) + \frac{S \cdot a}{2 \cdot \epsilon} \log\left(1 + e^{\pi/2}\right) \\
&\leq e^{\pi/2 + \epsilon - b} + \frac{S \cdot a}{2 \cdot \epsilon} e^{\pi/2} \\
&= e^{\pi/2} \cdot \left( e^{\epsilon - b} + \frac{S \cdot a}{2 \cdot \epsilon} \right) \,.
\end{aligned}
$$

$\square$

*Proof of Thm 5.2.* In Thm 5.1 we showed that the PGD loss can be taken arbitrarily close to zero. This implies that the PGD accuracy can be taken to 100%. Then, it is enough to show that $x_1 + a$ and $x_2 - 1$ are misclassified for CO to happen. To do so, we show that $f_{\boldsymbol{\theta}}(x_1 + a) \geq 0$ and $f_{\boldsymbol{\theta}}(x_2 - a) \leq 0$:

$$
\begin{aligned}
f_{\boldsymbol{\theta}}(x_1 + a) &= -h(-x_1 - a; \pi/2, a, b, \pi/2) \\
&= \mathrm{ReLU}(\pi/2 - a) - (1 + b/a) \cdot 0 + (1 + b/a) \cdot 0 \\
&= \pi/2 - a \geq 0 \text{ for } a \leq \pi/2 \,, \\
f_{\boldsymbol{\theta}}(x_2 - a) &= -h(-x_1 - a; \pi/2, a, b, \pi/2) \\
[\text{Eq. (25)}] &= -\pi/2 + a \leq 0 \text{ for } a \leq \pi/2 \,.
\end{aligned}
$$

Then, for $a \leq \pi/2$, we get the desired result. $\square$

*Proof of Thm 5.3.* Our goal is to upper bound:

$$
\frac{1}{n} \sum_{i=1}^{n} \mathop{\mathbb{E}}_{\delta_0^i \sim \mathrm{Unif.}([-\epsilon, \epsilon]^d)} \left[ \mathcal{L}(\boldsymbol{f_{\theta}}(\boldsymbol{x}_i + \boldsymbol{\delta}_S^i), y_i) \right] = \mathop{\mathbb{E}}_{\delta_0^1 \sim \mathrm{Unif.}([-\epsilon, \epsilon]^d)} \left[ \mathcal{L}(\boldsymbol{f_{\theta}}(\boldsymbol{x}_1 + \boldsymbol{\delta}_S^1), y_1) \right] \,,
\tag{26}
$$

which holds because $\boldsymbol{f_{\theta}}(\boldsymbol{x}_i + \boldsymbol{\delta})_{y_i} = \boldsymbol{f_{\theta}}(\boldsymbol{x}_j + \boldsymbol{\delta})_{y_j} \quad \forall \boldsymbol{\delta} \in [-\epsilon, \epsilon]^d, i, j \in [n]$ by construction of Eq. (8). We can further simplify by exploiting that $\boldsymbol{f_{\theta}}$ is constant along the last $d - 1$ dimensions of the input.

$$
\frac{1}{n} \sum_{i=1}^{n} \mathop{\mathbb{E}}_{\delta_0^i \sim \mathrm{Unif.}([-\epsilon, \epsilon]^d)} \left[ \mathcal{L}(\boldsymbol{f_{\theta}}(\boldsymbol{x}_i + \boldsymbol{\delta}_S^i), y_i) \right] = \mathop{\mathbb{E}}_{\delta_{01}^1 \sim \mathrm{Unif.}([-\epsilon, \epsilon])} \left[ \mathcal{L}(\boldsymbol{e}_{y_1} \cdot h(x_{11} + \delta_{S1}^1; x_{11}, a, b, \epsilon), y_1) \right]
$$

Finally, by noting that the output of wrong classes is always zero, i.e., $f_{\boldsymbol{\theta}}(\boldsymbol{x}_i + \boldsymbol{\delta})_j = 0 \ \ \forall \boldsymbol{\delta} \in [-\epsilon, \epsilon]^d, i \in [n], j \neq y_i$, we have that for all $\boldsymbol{\delta} \in [-\epsilon, \epsilon]^d$:

$$\mathcal{L}(f_{\boldsymbol{\theta}}(\boldsymbol{x}_1 + \boldsymbol{\delta}), y_1) = -\log\left(\frac{e^{f_{\boldsymbol{\theta}}(\boldsymbol{x}_1+\boldsymbol{\delta})_{y_1}}}{\sum_{j=1}^{o} e^{f_{\boldsymbol{\theta}}(\boldsymbol{x}_1+\boldsymbol{\delta})_j}}\right)$$

$$[-\log(a/b) = \log(b/a)] = \log\left(\frac{\sum_{j=1}^{o} e^{f_{\boldsymbol{\theta}}(\boldsymbol{x}_1+\boldsymbol{\delta})_j}}{e^{f_{\boldsymbol{\theta}}(\boldsymbol{x}_1+\boldsymbol{\delta})_{y_1}}}\right)$$

$$[\text{Zero score for wrong classes}] = \log\left(\frac{(o-1) + e^{f_{\boldsymbol{\theta}}(\boldsymbol{x}_1+\boldsymbol{\delta})_{y_1}}}{e^{f_{\boldsymbol{\theta}}(\boldsymbol{x}_1+\boldsymbol{\delta})_{y_1}}}\right) \quad (27)$$

$$= \log\left(1 + \frac{(o-1)}{e^{f_{\boldsymbol{\theta}}(\boldsymbol{x}_1+\boldsymbol{\delta})_{y_1}}}\right)$$

$$= \log\left(1 + (o-1) \cdot e^{-f_{\boldsymbol{\theta}}(\boldsymbol{x}_1+\boldsymbol{\delta})_{y_1}}\right)$$

$$[\text{Eq. (8)}] = \log\left(1 + (o-1) \cdot e^{-h(x_{11}+\delta_{S1}^1; x_{11}, a, b, \epsilon)}\right).$$

We can observe that this result resembles the loss in the proof of Thm 5.1, i.e., $\log\left(1 + e^{-h(x_2+\delta_S^2; x_2, a, b, \epsilon)}\right)$. In fact, for $o = 2$ classes they are equal. Since the sign of the gradient does not change, i.e., $\text{sign}\left(\frac{d}{d\delta}\log\left(1 + e^{-h(x_2+\delta; x_2, a, b, \epsilon)}\right)\right) = \text{sign}\left(\frac{d}{d\delta}\log\left(1 + (o-1) \cdot e^{-h(x_2+\delta; x_2, a, b, \epsilon)}\right)\right) \ \ \forall \delta \in [-\epsilon, \epsilon]$, we can re-use the argument employed in the proof of Thm 5.1 to upper bound the total loss, see Fig. 11. Then, Eq. (26) becomes:

$$\frac{1}{n}\sum_{i=1}^{n} \mathop{\mathbb{E}}_{\boldsymbol{\delta}_0^i \sim \text{Unif.}([-\epsilon,\epsilon]^d)}\left[\mathcal{L}(f_{\boldsymbol{\theta}}(\boldsymbol{x}_i + \boldsymbol{\delta}_S^i), y_i)\right] \leq P\left(\delta_{01}^1 \in \mathcal{D}_{\text{Low}}\right) \cdot \mathcal{L}(f_{\boldsymbol{\theta}}(\boldsymbol{x}_1 + \epsilon \cdot \boldsymbol{e}_1), y_1).$$

$$+ \left(1 - P\left(\delta_{01}^1 \in \mathcal{D}_{\text{Low}}\right)\right) \cdot \mathcal{L}(f_{\boldsymbol{\theta}}(\boldsymbol{x}_1 - a \cdot \boldsymbol{e}_1), y_1).$$

Now, we can just compute $\mathcal{L}(f_{\boldsymbol{\theta}}(\boldsymbol{x}_1 + \epsilon \cdot \boldsymbol{e}_1), y_1)$ and $\mathcal{L}(f_{\boldsymbol{\theta}}(\boldsymbol{x}_1 - a \cdot \boldsymbol{e}_1), y_1)$ thanks to Eq. (27), resulting in:

$$\mathcal{L}(f_{\boldsymbol{\theta}}(\boldsymbol{x}_1 + \epsilon \cdot \boldsymbol{e}_1), y_1) = \log\left(1 + (o-1) \cdot e^{-h(x_{11}+\epsilon; x_{11}, a, b, \epsilon)}\right)$$

$$[\text{Substituting on Eq. (7)}] = \log\left(1 + (o-1) \cdot e^{2\cdot\epsilon-a-b}\right)$$

$$= \log\left(1 + e^{\log(o-1)+2\cdot\epsilon-a-b}\right)$$

$$\leq e^{\log(o-1)+2\cdot\epsilon-b} = (o-1) \cdot e^{2\cdot\epsilon-b},$$

$$\mathcal{L}(f_{\boldsymbol{\theta}}(\boldsymbol{x}_1 - a \cdot \boldsymbol{e}_1), y_1) = \log\left(1 + (o-1) \cdot e^{-h(x_{11}-a; x_{11}, a, b, \epsilon)}\right)$$

$$[\text{Substituting on Eq. (7)}] = \log\left(1 + (o-1) \cdot e^{\epsilon-a}\right)$$

$$= \log\left(1 + e^{\log(o-1)+\epsilon-a}\right)$$

$$\leq e^{\log(o-1)+\epsilon} = (o-1) \cdot e^{\epsilon}.$$

Putting everything together, we obtain the desired result:

$$\frac{1}{n}\sum_{i=1}^{n} \mathop{\mathbb{E}}_{\boldsymbol{\delta}_0^i \sim \text{Unif.}([-\epsilon,\epsilon]^d)}\left[\mathcal{L}(f_{\boldsymbol{\theta}}(\boldsymbol{x}_i + \boldsymbol{\delta}_S^i), y_i)\right] \leq \frac{2\cdot\epsilon - S\cdot a}{2\cdot\epsilon} \cdot (o-1) \cdot e^{2\cdot\epsilon-b} + \frac{S\cdot a}{2\cdot\epsilon} \cdot (o-1) \cdot e^{\epsilon}$$

$$\leq (o-1) \cdot e^{2\cdot\epsilon} \cdot \left(e^{-b} + \frac{S\cdot a}{2\cdot\epsilon}\right).$$

$\square$

*Proof of Thm 5.4.* Similarly to the proof of Thm 5.2, it is enough to show that the points $\boldsymbol{x}_i - a \cdot \boldsymbol{e}_1$ are misclassified. To do so, we will show that the logit for a class different than $y_i$, which is zero, is larger than

the logit of the class $y_i$:

$$f_{\boldsymbol{\theta}}(\boldsymbol{x}_i - a \cdot \boldsymbol{e}_1)_{y_i} = h(x_{i1} - a; x_{i1}, a, b, \epsilon)$$
$$[\text{Substituting on Eq. (7)}] = a - \epsilon.$$

Since we have that $a \leq \epsilon/S$ and $S \geq 1$, we have that $f_{\boldsymbol{\theta}}(\boldsymbol{x}_i - a \cdot \boldsymbol{e}_1)_{y_i} \leq 0$ for all $i \in [n]$, meaning that the true adversarial accuracy is zero and therefore, according to Thm 2.1, CO exists. $\qquad\square$

## C  Additional experimental validation

### C.1  Experimental setup

For the image classification experiments, we use the popular MNIST (LeCun et al., 1998), SVHN (Netzer et al., 2011) and CIFAR10 (Krizhevsky, 2009) datasets. We train our models with Stochastic Gradient Descent, momentum 0.9, batch size 128 and the standard schedule with cyclic learning rate proposed by Andriushchenko & Flammarion (2020), with 15 epochs for MNIST/SVHN and 30 epochs for CIFAR10. We employ the PreActResNet18 architecture (He et al., 2016) and $\sigma = 0$ in Alg. 1 in all of our image classification experiments. For different learning rate schedules like Rice et al. (2020) and architectures like ViT-Small (Dosovitskiy et al., 2021), we refer to Sec. C. For Sec. 6.2, we employ the standard weight decay 0.0005. All of our training setups are repeated over 3 random seeds to report the average performance and confidence. All of our experiments are conducted on a single machine with an NVIDIA A100 SXM4 40 GB GPU. Our codebase is based on torch (Paszke et al., 2017) and timm (Wightman, 2019).

### C.2  The effect of weight decay

In this section, we present our $\lambda_{\text{wd}}$ selection and SVHN results. We test $\lambda_{\text{wd}} \in \{0, 0.0005, 0.001, 0.005, 0.01, 0.05\}$ for CIFAR10 and $\lambda_{\text{wd}} \in \{0, 0.0005, 0.001, 0.005, 0.01, 0.0125, 0.05\}$ for SVHN. In order to evaluate the $\lambda_{\text{wd}}$ values, we evaluate the PGD-20 Acc. in a validation set of 1024 samples extracted from the training set. Then, we select the values that do not present CO.

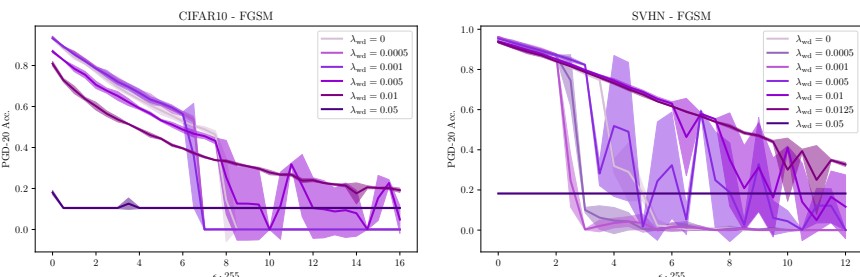

Figure 12: $\lambda_{\text{wd}}$ **selection for CIFAR10 and SVHN:** Small values of lambda present CO at different $\epsilon_c$. The largest value $\lambda_{\text{wd}} = 0.05$ presents the performance of a constant classifier predicting the most numerous class, i.e., $\sim 10\%$ Acc. for CIFAR10 and $\sim 20\%$ Acc. for SVHN. Finally, $\lambda_{\text{wd}} = 0.01$ and $\lambda_{\text{wd}} = 0.0125$, for CIFAR10 and SVHN respectively, are able to avoid CO and present non-zero PGD-20 Acc. for every $\epsilon$.

In Fig. 12 we can observe that the only values that do not present CO and do not converge to a constant classifier are $\lambda_{\text{wd}} = 0.01$ for CIFAR10 and $\lambda_{\text{wd}} = 0.0125$, for SVHN. In Fig. 13 we can observe that the performance with $\lambda_{\text{wd}} = 0.0125$ is less than with ELLE-A. However, the gap is not as pronounced as in CIFAR10, see Fig. 6.

We additionally compare the performance between the models when using $\lambda_{wd} = 0.0005$ and $\lambda_{wd} = 0$ in SVHN for all $\epsilon$ and $S \in [1, 2, 3]$. In Fig. 14, we can observe that the phase transition is observed for larger $\epsilon$ when employing $\lambda_{\text{wd}} = 0$. Since the rest of training hyperparameters were optimized for $\lambda_{\text{wd}} = 0.0005$ by Andriushchenko & Flammarion (2020), training does not converge for $\lambda_{\text{wd}} = 0$, explaining this phenomenon.

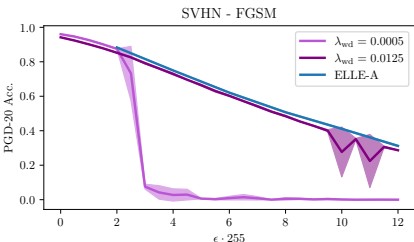

Figure 13: **Weight decay and local linearity regularization on SVHN:** We can observe that $\lambda_{\text{wd}} = 0.0125$ avoids CO, but the performance is worse than of ELLE-A. This means that local linearity regularization is less restrictive to the classifier while avoiding CO.

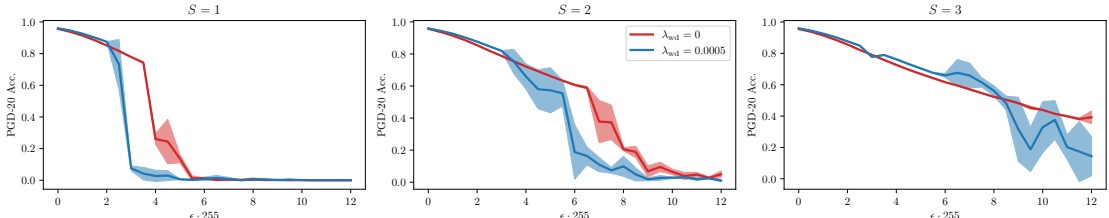

Figure 14: **Phase transition with and without weight decay in SVHN:** We train with the original $\lambda_{\text{wd}} = 0.0005$ employed by Andriushchenko & Flammarion (2020) and $\lambda_{\text{wd}} = 0$. We observe that the phase transition occurs for larger $\epsilon$ when employing $\lambda_{\text{wd}} = 0$. This is due to not converging with $\lambda_{\text{wd}} = 0$, which is a result of the rest of training hyperparameters being optimized for $\lambda_{\text{wd}} = 0.0005$ by Andriushchenko & Flammarion (2020).

### C.3  Re-scaling for obtaining smaller $\epsilon_c$

in Thm C.1 we present a mechanism that can induce arbitrarily small $\epsilon_c$ for which CO appears in more complex models, where the only assumption is that the first layer is affine.

**Proposition C.1** (Re-scaling the data re-scales $\epsilon$). *Given a classification dataset $\{(\boldsymbol{x}_i, y_i)\}_{i=1}^n$ and a model $\boldsymbol{f}_{\boldsymbol{\theta}} : \mathbb{R}^d \to \mathbb{R}^o$ where the first layer is affine, i.e., $\boldsymbol{f}_{\boldsymbol{\theta}}(\boldsymbol{x}) = \hat{\boldsymbol{f}}_{\hat{\boldsymbol{\theta}}}(\boldsymbol{W}\boldsymbol{x} + \boldsymbol{b})$, where $\hat{\boldsymbol{f}}_{\hat{\boldsymbol{\theta}}}$ are all the layers except the first one and $\boldsymbol{\theta} = \hat{\boldsymbol{\theta}} \cup \{\boldsymbol{W}, \boldsymbol{b}\}$. Let $\alpha \in \mathbb{R} \setminus \{0\}$. Solving Eq. (AT) in a re-scaled dataset $\{(\alpha \cdot \boldsymbol{x}_i, y_i)\}_{i=1}^n$ with adversarial budget $\epsilon_\alpha$ is equivalent to solving Eq. (AT) in the standard dataset $\{(\boldsymbol{x}_i, y_i)\}_{i=1}^n$ with adversarial budget $\epsilon = \epsilon_\alpha / \alpha$.*

**Corollary C.2** (Re-scaled datasets present re-scaled $\epsilon_c$). *If training with Alg. 1 in the dataset $\{(\boldsymbol{x}_i, y_i)\}_{i=1}^n$ presents a critical adversarial budget $\epsilon_c$ as in Thm 2.2, training in the re-scaled dataset $\{(\boldsymbol{x}_i \cdot \alpha, y_i)\}_{i=1}^n$ will present a critical adversarial budget $\epsilon_{c,\alpha} = \alpha \cdot \epsilon_c$.*

With Thms C.1 and C.2, we have a mechanism to re-scale the dataset and produce smaller $\epsilon_c$ that applies to modern deep architectures like ResNets (He et al., 2016) and any training dataset. In particular, Thm C.2 shows that the scale of $\epsilon_c$ does not only depend on the hyperparameters of Alg. 1, but on the scale of the dataset. In Sec. C.3 we confirm the result holds for ResNets trained in MNIST, SVHN and CIFAR10.

In order to empirically validate Thms C.1 and C.2, we train in the modified dataset $\{(\boldsymbol{x}_i \cdot \alpha, y_i)_{i=1}^n\}$ with $\alpha \in \{0.25, 0.5, 0.75\}$, with $\alpha = 1$ being the original dataset. First, we train in MNIST, SVHN and CIFAR10 with $S = 1$. Then, we focus in the SVHN dataset, where CO can happen for $S \in \{1, 2, 3\}$ (See Fig. 5) in order to better analyze how the critical values of epsilon are displaced for larger $S$.

In Fig. 15, we can observe a clear proportionality in $\epsilon_c$ with respect to $\alpha$, with CO appearing earlier the smaller the $\alpha$. Similarly, for any number of steps $S$ for SVHN, the critical values $\epsilon_c$ are re-scaled accordingly, confirming the result in Thm C.2 and showing that CO can be induced for small $\epsilon$.

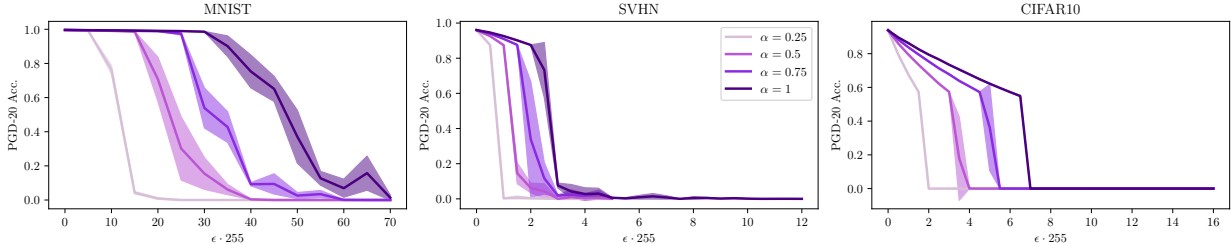

Figure 15: **Phase transition in re-scaled datasets with FGSM:** We preprocess the datasets by multiplying the inputs $\boldsymbol{x}$ by $\alpha \in \{0.25, 0.5, 0.75, 1\}$. Re-scaling the dataset inputs by a factor of $\alpha$ produces proportionally smaller $\epsilon_c$ in single-step AT.

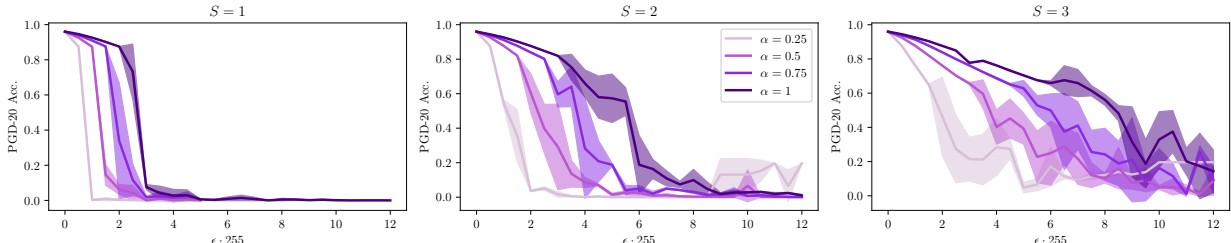

Figure 16: **Phase transition in re-scaled SVHN with multiple steps:** We preprocess the datasets by multiplying the inputs $\boldsymbol{x}$ by $\alpha \in \{0.25, 0.5, 0.75, 1\}$ and train with $S \in \{1, 2, 4\}$ PGD steps. Even with larger $S$, CO can be produced earlier by re-scaling the training dataset.

### C.4 The phase transition with longer schedules

Recent works argue longer training schedules might lead to CO (Kim et al., 2021; Abad Rocamora et al., 2024). According to our analysis this can be the case, as longer schedules might converge to the CO solutions shorter schedules did not. In Fig. 17 we find that $\epsilon_c$ is slightly larger for MNIST and SVHN and slightly smaller for CIFAR10.

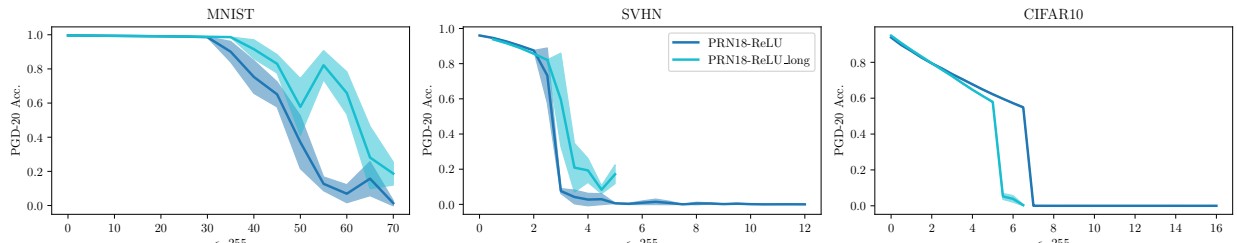

Figure 17: **Phase transition with a** 30 **or** 15 **epoch** ● **v.s. a** 200 **epoch** ● **schedule:** The phase transition occurred later for the studied $\epsilon$ in the long schedule for MNIST and SVHN, on the contrary, the phase transition occurred 1.5/255 points earlier with the long schedule in CIFAR10.

### C.5 The effect of architecture in the phase transition

Singla et al. (2021) argue low-curvature architectures like ResNets with the Swish activation provide better properties regarding robust overfitting. In this experiment, we evaluate the appearance of CO for PreActResNet with both the ReLU and Swish activation. We additionally evaluate the performance of ViT-small (Dosovitskiy et al., 2021), which we train with the proposed training hyper-parameters in Wu et al. (2022) (Sec. 5.1). We used an embedding size of 384, a patch size of 4, and 6 heads.

In Fig. 18, we can observe CO occurs later for ViT-small and PRN18-Swish, confirming the result of Singla et al. (2021) for the swish activation.

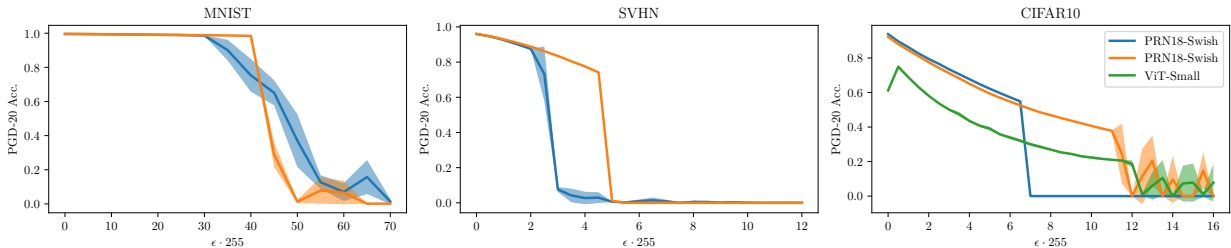

Figure 18: **Phase transition with PRN18-ReLU ●, PRN18-Swish ● and ViT-small ●:** The phase transition occurs later for ViT-small and PRN18-Swish, implying that architecture plays a role in the appearance of CO.

### C.6  Visualizing the loss landscape

In this section, we visualize the loss landscape of the learnt models in the CIFAR10 dataset with $S = 1$ and $\epsilon \in \{4, 16\}/255$ and $S = 3$ with $\epsilon = 16/255$. Following Kim et al. (2021); de Jorge et al. (2022), in Table 2 we plot the loss values for the points $\boldsymbol{x}_{i,\alpha,\beta} = \boldsymbol{x}_i + \alpha \cdot \boldsymbol{\delta}_{\text{FGSM}} + \beta \cdot \boldsymbol{\delta}_{\text{rand}}$, where $\boldsymbol{\delta}_{\text{rand}}$ is a random vector satisfying $||\boldsymbol{\delta}_{\text{rand}}||_\infty = \epsilon$ and $\alpha \in [-1, 1]$, $\beta \in [-1, 1]$. We take the first 4 samples of the CIFAR10 dataset. For $S = 1$ at $\epsilon = 4/255$ and $S = 3$ at $\epsilon = 16/255$ no CO is observed and the loss landscale is locally linear. When CO appears, i.e., $S = 1$ and $\epsilon = 16/255$, the local linearity of the loss landscape is lost, following the findings of Andriushchenko & Flammarion (2020). Moreover, the loss at $\boldsymbol{x}_i + \boldsymbol{\delta}_{\text{FGSM}}$ is lower than the loss at points with $\alpha \approx 0$ and $\beta \approx 0$, where the maximum of the loss is achieved for all the points.

Table 2: **Visualization of the loss landscape with and without CO:** We plot the loss values for the PreActResNet18 models trained in CIFAR10 for the first 4 training samples. We paint the regions where the predictions are correct and not correct in blue and red respectively. We observe that FGSM with $\epsilon = 16/255$, which presents CO, has a distorted landscape with one or more maxima near $\boldsymbol{x}_i$ and low loss at $\boldsymbol{x}_i + \boldsymbol{\delta}_{\mathrm{FGSM}}$. In the other cases, CO does not appear and the loss landscapes remain locally linear, as shown by Andriushchenko & Flammarion (2020).

| Training ID ($i$) | FGSM - $\epsilon = 4/255$ | FGSM - $\epsilon = 16/255$ | AT-3 - $\epsilon = 16/255$ |
|---|---|---|---|

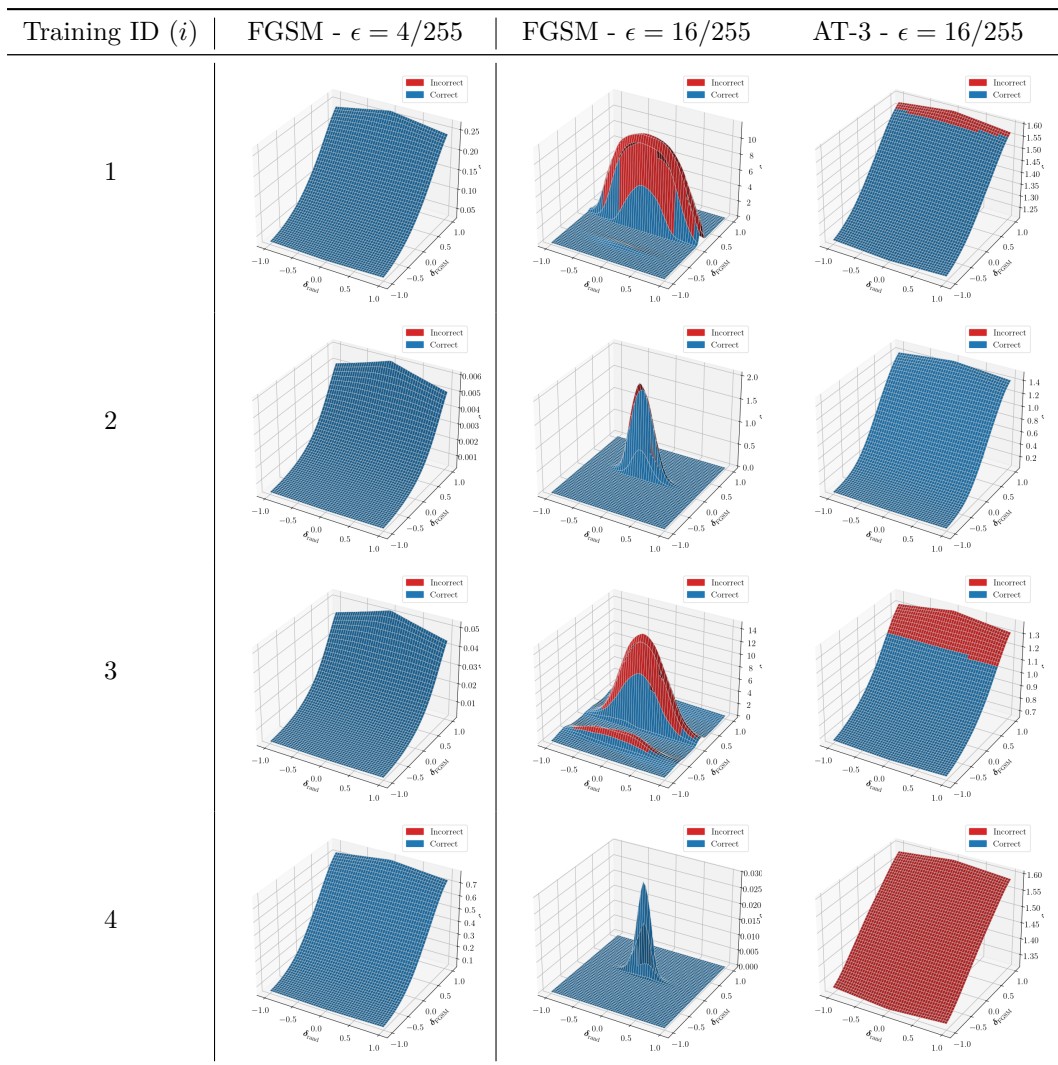

