# OpenReview forum: "A Mechanistic View of Catastrophic Overfitting"
_TMLR — Rejected by TMLR_

### Review · Reviewer_QbQr · 2026-02-26

**Summary Of Contributions:**

The paper attempts to explain catastrophic overfitting, a phenomena which occurs during single step adversarial training. CO is characterized by a sudden drop in robust accuracy against more complex adversaries than the ones seen during training. The authors conduct a simple toy experiment and relate CO to a phase transition. Further, they try to investigate the relationship between the number of PGD steps and PGD initialization and catastrophic overfitting (which should than be called robust overfitting).

**Audience:**

No

**Audience Explanation:**

The manuscript would benefit from a more rigorous theoretical grounding. Specifically, the definition of catastrophic overfitting remains imprecise, and the paper fails to distinguish this phenomenon from related concepts, such as robust overfitting. While the observations regarding overfitting are noted, the generalizability of the claims is limited. The current evaluation does not incorporate state-of-the-art models or standard benchmark datasets, which is necessary to substantiate the authors' broader conclusions. As it stands, the final synthesis remains too vague to offer actionable insights for the field.

**Broader Impact Concerns:**

There is a concern regarding the scientific signaling of this work. Given that the core claims lack sufficient empirical support and rely on imprecise definitions, formal publication in its current state may inadvertently promote unverified methodologies.

**Claims And Evidence:**

No

**Claims Explanation:**

The authors mix many concepts into one paper and do not fully get prior work. Further, the claims made are not applicable for large scale networks or datasets. Their toy model experiments do not transfer and e.g. Figure 5 does not show a phase transition but rather simple catastrophic overfitting, due to a more complex adversary during testing than during training.

**Requested Changes:**

In its current state, the manuscript is not suitable for publication due to several foundational conceptual flaws and a lack of empirical rigor.

Further comments:

-	The introduced variable $\epsilon_c$ is not clear, and the stated reference also does not contain a helpful explanation.
-	Figure 2 is intended to present a phase transition, but it is not clear what it is actually showcasing. What I see is catastrophic overfitting plotted as FGSM loss vs. AT loss, where it is not clear which adversarial training method the model was trained with.
-	It is not clear why the findings on the toy model should translate to larger models.
-	The abbreviation "Thms" is used but never introduced; I assume it stands for "Theorems."
-	In Figure 5, some lines are missing; it simply plots catastrophic overfitting, and I do not see how this showcases a phase transition.

---

> ### Author Response · Authors · 2026-03-02
> **Thanks for your review**
>
> Dear Reviewer QbQr,
>
> Thanks for your time reviewing our paper. We reply to your comments as follows:
>
> **P1: "authors mix many concepts into one paper and do not fully get prior work"**
>
> We are unsure about which of our statements/contributions conflicts with prior work. Could you please clarify which prior work is not accurately portrayed in our work?
>
> **P2: "the claims made are not applicable for large scale networks or datasets"**
>
> We point to Theorem 5.3, where we construct solutions with arbitrarily small (near-zero) loss for PGD attacks but guaranteed adversarial examples. This result holds for **any well-separated dataset**. MNIST, CIFAR10 or SVHN satisfy this condition as shown by [1]. The implication of this result is that a powerful neural network can approximate solutions like those in Theorem 5.3 and result in CO. As stated at the end of Section 5.2, this agrees with experimental observations in other works regarding CO appearing with random initialization or larger model sizes.
>
> **P3: "Figure 5 does not show a phase transition but rather simple catastrophic overfitting"**
>
> One contribution of this work is to associate CO with a phase transition. Intuitively, a phase transition is a non-smooth change in a certain metric. For us, the metric we are interested in analyzing is the robust accuracy / loss. Therefore, in our case, a phase transition is observed as a sudden increase in the robust loss and decrease in the robust accuracy for a small change in $\epsilon$.
>
> In the toy model (Section 3), we are able to fully characterize the phase transition and for which value of $\epsilon$ it appears. In Figure 5, the observations are empirical. There are two key differences with respect to the analysis in the toy model: 1) In Figure 5 we plot the 20-step PGD accuracy, which is an upper bound of the robust accuracy. 2) In Figure 5, for each $\epsilon$ value, we report the performance obtained after training with SGD, not the best possible solution. However, the phase transitions are clearly observable as sudden decreases in PGD-20 accuracy.
>
> **P4: "the definition of catastrophic overfitting remains imprecise, and the paper fails to distinguish this phenomenon from related concepts, such as robust overfitting"**
>
> Robust overfitting is often discussed in terms of the training dynamics [3], i.e. how testing and training performance evolve epoch after epoch. Similarly to "standard" overfitting, after reaching the best test performance at a certain epoch, the adversarial accuracy in the training set keeps improving at the cost of degrading the adversarial accuracy in the test set. A connection could be made between robust and catastrophic overfitting arguing that catastrophic overfitting is an extreme case of robust overfitting where the test adversarial accuracy drops to zero. A key difference is that CO also happens within the training set, where the FGSM accuracy grows and the PGD accuracy drops to zero, see Figure 2 in [4]. This is captured by Definition 2.1. We are happy to add a discussion comparing robust and catastrophic overfitting and comment that $\eta$ should be close to zero to make the difference even clearer.
>
> **P5: "The current evaluation does not incorporate state-of-the-art models or standard benchmark datasets"**
>
> We incorporate standard models and datasets tested in the community, e.g., PreActResNet18 and CIFAR10. Additional architectures such as ViT are evaluated in Appendix C.5.
>
> **P6: "the final synthesis remains too vague to offer actionable insights for the field"**
>
> As stated in the introduction, the problem we approach is understanding why CO happens and the role of the number of PGD steps $S$ and random initialization on its appearance. We do not aim at offering a solution to CO, but at understanding its source.
>
> **P7: "The introduced variable $\epsilon_c$ is not clear, and the stated reference also does not contain a helpful explanation."**
>
> We point to Definition 2.2 for a description of $\epsilon_c$. The intuition is that $\epsilon_c$ captures the last $\epsilon$ before CO appears. For example, in CIFAR10 with FGSM ($S=1$), we have that $\epsilon_c = 13/510$ (see Figure 5). We will be happy to clarify any part of Definition 2.2 you think can be improved.

---

> ### Author Response · Authors · 2026-03-02
>
> **P8: "Figure 2 is intended to present a phase transition, but it is not clear what it is actually showcasing. What I see is catastrophic overfitting plotted as FGSM loss vs. AT loss, where it is not clear which adversarial training method the model was trained with."**
>
> We refer to Section 6.2 for details on this experiment. Summarizing, **there is no training involved**, since there is a single parameter to optimize, we evaluate the FGSM and AT losses over 10,000 evenly-spaced values over the [0,10] interval and get the one with the minimum loss. This approximates the global minimum. Similarly, to approximate the AT loss (best possible adversary), we evaluate the loss over 100 evenly-spaced values of the adversarial perturbation and get the one with maximum loss. The phase transition occurs when the AT loss spikes up separating from the FGSM loss at $\epsilon_c \approx \pi/8$.
>
> **P9: "It is not clear why the findings on the toy model should translate to larger models."**
>
> We assume you refer to the results in Theorem 4.1 and its corollaries. Essentially, the main result is that CO can happen for any number of steps $S$ and arbitrarily small $\epsilon$. While the result for arbitrarily small $\epsilon$ is only observed in the toy model, CO is in practice observed in larger models for $S>1$ as observed in Figure 5 and by [5]. Regarding the effect of the parameter norm in CO, the exact dependence on $|\theta|$ and $S$ described in Corollary 5.3 is specific to the toy model. However, as studied in Section 6.3, the intuition that penalizing parameter norms helps avoid CO also holds for larger models like PreActResNet18 in CIFAR10 and SVHN.
>
> **P10: "The abbreviation "Thms" is used but never introduced; I assume it stands for "Theorems.""**
>
> Yes, thanks for pointing this out, we will clarify it in the revised version of the manuscript.
>
> **P11: "In Figure 5, some lines are missing; it simply plots catastrophic overfitting, and I do not see how this showcases a phase transition."**
>
> Please note that there are no lines missing, in Section 6.2 we mention that "For each $S$, we start training at the first $\epsilon$ value that presented CO for $S − 1$ steps".
>
> **P12: "formal publication in its current state may inadvertently promote unverified methodologies."**
>
> We respectfully disagree with this comment. Our theoretical results are valid and intuitive to understand with visualizations (Figures 1 to 4), they showcase how CO appears as a phase transition and why adding multiple PGD steps during training, or adding random initialization cannot avoid CO in these theoretical models. These theoretical insights bring a novel view of CO to the community.
>
> Moreover, our theoretical insights are in line with previous empirical observations of CO with multiple PGD steps [5] or with larger models [6]. Our theoretical insights are also tested in practical scenarios where CO is observed, e.g., PreActResNet18 in CIFAR10, where we show that weight-decay regularization can avoid CO as motivated by Corollary 4.3.
>
> Thanks again for your time. We would appreciate follow-up comments on our clarifications. We remain available in case further questions arise.
>
> **References**
>
> [1] Yang et al., A Closer Look at Accuracy vs. Robustness. NeurIPS 2020
>
> [2] Wong et al., Fast is better than free: Revisiting adversarial training. ICLR 2020
>
> [3] Rice et al., Overfitting in adversarially robust deep learning. ICML 2020
>
> [4] Kim et al., Understanding Catastrophic Overfitting in Single-step Adversarial Training. AAAI 2021
>
> [5] He et al.,  Investigating catastrophic overfitting in fast adversarial training: A self-fitting perspective. arXiv 2023
>
> [6] Abad Rocamora et al., Efficient local linearity regularization to overcome catastrophic overfitting. ICLR 2024

---

### Review · Reviewer_AGe8 · 2026-03-15

**Summary Of Contributions:**

### Summary
This paper makes contributions to the theoretical foundation of catastrophic overfitting. Specifically, the paper interprets the catastrophic overfitting as a phase transition in the adversarial budget. To observe this phase transition, the paper starts with a simple, mathematically tractable model, then shows that there is a critical transition point $\epsilon_c$ such that, if the adversarial budget is larger than $\epsilon_c$, the optimal solution from adversarial training with FGSM/PGD differs from the theoretical optimal solution. In particular, for the specific model choice, this transition point can be set arbitrarily small, regardless of the number of PGD steps. The paper tries to generalize this phase transition to more practical settings, e.g., ReLU activation, and practical datasets. The paper also presents a few experiments demonstrating the phase transition phenomenon and a potential strategy to prevent catastrophic overfitting.

### Strengths
1. The phase transition hypothesis is interesting and worth more investigation.
2. The hypothesis might imply some tricks to prevent catastrophic overfitting.

### Weaknesses
1. Many findings are limited to the “mathematically tractable” models. I don’t think that the paper succeeded in generalizing the observation.
2. The model choice is very unusual to reflect the practice. This choice should be justified in the paper, but there is no explanation of why it is suitable.
3. Even the ReLU generalization adds unusual assumptions on the model parameters. In practice, there is no guarantee that the parameters are constrained in a specific way, so the observed phase transition would not generalize to most NNs. Because Section 5.2 also resorts to Equation 7 derived from this specific setting, I disagree that the paper generalized the finding to the practical setting.
4. There are too few experiments supporting the paper’s finding. I don’t think the paper provided sufficient theoretical evidence for the claimed phase transition in practice, so I expect more extensive experiments to demonstrate it.

**Additional Comments:**

This paper contains an interesting theory about the root cause of the catastrophic overfitting. In general, the paper is clear and easy to understand. However, the paper’s findings are very limited to a specific setting, and it lacks sufficient evidence of generalization to more practical settings.

**Audience:**

Yes

**Audience Explanation:**

Generally, both researchers and practitioners are interested in theoretical explanations of the causes of catastrophic overfitting. If the phase transition can be (formally and practically) demonstrated, the phenomenon warrants further (both theoretical and experimental) investigation.

**Broader Impact Concerns:**

I don’t see a particular broader impact concern regarding this paper.

**Claims And Evidence:**

No

**Claims Explanation:**

As noted in the Weaknesses section, this paper examines the phase transition in a very specific, mathematically tractable model. The choice of model is too unusual to reflect typical model training practice, so Sections 3 and 4 do not provide any evidence that a phase transition should exist in a practical setting, e.g., training a deep neural network with FGSM-AT. I disagree that Section 5 generalizes the observation sufficiently, because the paper again makes a very unusual assumption about the model’s parameters. There are too few experiments in Section 6, so I don’t see enough evidence that the phase transition is a universal phenomenon that can explain the catastrophic overfitting.

**Requested Changes:**

1. While the toy model in Sections 3 and 4 is illustrative, the model does not reflect the models in practice. Given this gap between the toy model and practice, those sections take up too many pages in the main body of the paper. Consider rewriting the section to contain minimal discussions to illustrate the findings.
2. To generalize the finding, generalizing the observation to the ReLU activation is essential, and there should not be a limitation that seems far from practice.
3. Section 6 should be improved further.
    * Experimental setup should be briefly described in the main body of the paper, rather than deferred to the Appendix.
    * Section 6.1 does not contain any new experimental findings. This section describes the experimental explorations in the previous sections, particularly Figures 1 and 2. Just move the discussions to the previous sections.
    * The experiment in Section 6.2 should be done more extensively. The experiments used only one model and 3 random seeds, and this would not cover many different scenarios in practice. Please do more experiments with various models with many different seed values.
    * Section 6.3 contains a meaningful insight in practice, and more experiments should be added to support this insight.

---

> ### Author Response · Authors · 2026-03-25
>
> Dear Reviewer AGe8,
>
> Thanks for reviewing our work. Let us reply to your comments:
>
> **P1: "Many findings are limited to the “mathematically tractable” models. I don’t think that the paper succeeded in generalizing the observation."**
>
> Thanks for highlighting this point. We believe you refer to the appearance of CO due to a phase transition.
>
> The motivation behind studying our mathematically-tractable models, is to find the simplest setup where CO can be reproduced with the same observations as in larger scales, but with a complete understanding of the mechanisms behind it. Importantly, through our small-scale models, we are the first to fully characterize the mechanism leading to CO.
>
> As argued in Section 7 (Limitations and future work), at the moment, the phase transition can only be demonstrated empirically in large scale models. Nevertheless, the fact that the phase transition can be demonstrated in two different architectures: Sinusoidal (Section 3.2) and ReLU (Appendix A.2) models, together with empirical observations (in our work and past literature), suggests that the phase transition is not an artifact from small-scale models, but a real phenomenon leading to CO.
>
> **P2: "The model choice is very unusual to reflect the practice.", "generalizing the observation to the ReLU activation is essential"**
>
> Let us clarify why the sinusoidal model was chosen in the main text. We chose the sinusoidal model because it is differentiable everywhere and the FGSM loss function can be easily expressed as a closed formula (Equation 3). Let us classify that **the phase transition can also be demonstrated with ReLU models**. In Appendix A.2, we repeat the same analysis done in Section 3.2 with a ReLU-based model. The phase transition can also be demonstrated in this case. In Figure 10, for $\epsilon=0.1$, the optimal solution is the robust solution ($\theta=0$), while at $\epsilon=1$, the solution is the non-robust ($\theta=10$).
>
> **P3: "Even the ReLU generalization adds unusual assumptions on the model parameters... I disagree that the paper generalized the finding to the practical setting."**
>
> Thanks for highlighting this point. Let us clarify that **the result in Theorem 5.2 does not require constraining the weights**. Additionally, this result is not related to the phase transition result, but rather to the question **Q2: "What is the role of the number of PGD steps S and its initialization in the appearance of CO?""**. This result helps us answer **Q2** by showing that increasing the number of steps $S$ or adding random PGD initialization cannot avoid CO. This is our second contribution. Intuitively, Theorem 5.2 does not describe a model presenting CO, but a CO solution that models can converge to.
>
> **P4: "There are too few experiments supporting the paper’s finding.", "The experiment in Section 6.2 should be done more extensively.", "Please do more experiments with various models with many different seed values."**
>
> The main goal of our paper is not to demonstrate the empirical appearance of CO, but to understand its sources. We perform experiments to analyze theoretical insights arising from small-scale models. Mainly, the insights from Corollary 4.3 regarding constraining the model weights $\theta$ (Section 6.3) or augmenting the number of steps $S$ (Section 6.2).
>
> In Section 6.3, for CIFAR10 only, we trained models for 33 different $\epsilon$ values, 3 different seeds and 6 different weight decay parameters, totalling 594 training runs. A number of runs in the same order was performed for SVHN, we refer to the full experimental results in Appendix C.2. The results in this section consistently show that CO can be controlled by tuning the weight decay parameter, at a cost of degrading the final model performance.
>
> In Appendices C3, C4 and C5 we evaluate re-scaled datasets, longer training schedules and different architectures such as ViT. In all of these experiments, the sudden drop in multi-step PGD accuracy (characteristic of a phase transition) is observed. These results, together with previous observations in the literature, make us believe that a phase transition is the most likely explanation to CO.
>
> **P5: "Experimental setup should be briefly described in the main body of the paper, rather than deferred to the Appendix."**
>
> Thanks for this comment. We agree that adding the experimental setup in the main paper would add value to the paper. We will include this change in the revised version of the manuscript.
>
> **P6: "This section (6.1) describes the experimental explorations in the previous sections, particularly Figures 1 and 2. Just move the discussions to the previous sections."**
>
> Thanks for this suggestion. We will incorporate this change.
>
> We appreciate your feedback. Your suggestions will be incorporated in the revised version of the manuscript. Please let us know if follow-up questions appear, we will be happy to answer them.
>
> Best regards,
>
> Authors.

---

### Review · Reviewer_JK3i · 2026-03-18

**Summary Of Contributions:**

This paper studies catastrophic overfitting (CO) in adversarial training and aims to provide a mechanistic perspective on its appearance. The authors analyze CO using mathematically tractable models and interpret it as a phase transition phenomenon in adversarial budget $\epsilon$. The paper further examines roles of PGD steps and initialization, and argues that CO solutions can arise for well-separated datasets regardless of step number or random initialization. Empirical experiments on image classification datasets are also presented to support discussion.
The topic is interesting and relevant. Using simple analytical models to study CO is appealing and offers helpful intuition. The notion of critical adversarial budget is also conceptually interesting and may be useful for both theoretical and practical discussions.  However, several aspects of paper would benefit from clarification. In particular, notation and explanations are sometimes insufficient. Some central interpretations are not fully supported by analysis, and implications of theoretical results are not always clearly discussed beyond toy settings.

**Audience:**

Yes

**Audience Explanation:**

Researchers working on adversarial robustness and optimization would likely find this work interesting. Catastrophic overfitting is widely observed but still not fully understood. Mechanistic viewpoint and phase transition perspective may encourage further discussion and follow-up research. Concept of critical adversarial budget may also motivate new theoretical and empirical work.

**Broader Impact Concerns:**

This work is mainly theoretical and methodological. Better understanding of adversarial robustness may help development of more reliable machine learning systems. At the same time, insights into limitations of adversarial training could also inform stronger attacks. These concerns are common in robustness research, and no additional major risks appear specific to this paper.

**Claims And Evidence:**

No

**Claims Explanation:**

A main concern is about interpretation of theoretical results. Paper appears to frame contribution as explaining why CO occurs, but much of analysis mainly shows that CO can arise under certain constructions. Demonstrating existence of CO in simplified models is valuable, but it does not necessarily provide full explanation of mechanism in general settings. Clarifying this distinction would strengthen contribution.

Relatedly, CO can naturally occur when adversarial examples used during training differ from those used during evaluation. From this perspective, CO may be viewed as expected outcome rather than phenomenon requiring deeper explanation. If this interpretation is accepted, theoretical findings may be better framed as supporting conjecture that CO is unavoidable under certain conditions, rather than explaining its cause.

The notion of critical adversarial budget is particularly interesting, but its interpretation remains unclear. Paper suggests that CO appears only above threshold ϵc, yet discussion of why such threshold exists is limited. It would be helpful to provide more intuition or theoretical explanation. It would also be useful to clarify practical meaning. For example, it is not clear whether small training budget implies stability against CO, since training and testing attacks may use different budgets.

Another issue concerns scope of results. Many arguments rely on toy models or structured constructions. While authors suggest similar behavior in complex models, more discussion is needed on whether these findings are expected to generalize and what assumptions are required.

Finally, there are several issues related to notation and clarity:
  1) In Algorithm 1, it is unclear whether index $i$ refers to individual samples or mini-batches.
  2) In Section 3.1, cross-entropy loss is written as 𝓛(θ,ϵ), although it appears it should be written as 𝓛(θ).
  3) Equation (3) seems to require $\sum_i$.
  4) Corollaries 4.2 and 4.3 are labeled inconsistently in some places (e.g. Thm 4.2, Thm 4.3).

**Requested Changes:**

I would like to suggest a few revisions that may help improve clarity and strengthen overall contribution of the paper:

1. It would be helpful to clarify the main claim and positioning of the work. In particular, distinguishing more clearly between explaining mechanism of CO and demonstrating its existence in specific settings could make contribution easier to understand.

2. The concept of critical adversarial budget is very interesting. Providing more intuition or theoretical reasoning for why such a threshold arises would improve paper. In addition, a clearer discussion of its practical meaning would be valuable.

3. Expanding discussion on how insights from toy models relate to more complex models would strengthen impact. It would be useful to explain more explicitly what aspects are expected to generalize and what limitations remain.

4. In experiments on real datasets, discussion subjects on weight decay. It would be helpful if authors could further explain how this relates to boundedness condition in Corollary 4.3. Providing additional intuition on this connection may clarify interpretation of theoretical results. In addition, more detailed empirical analysis on real datasets would strengthen practical relevance of findings.

5. Improving notation and presentation would make paper easier to follow. Clarifying ambiguous parts and ensuring consistency in definitions, equations, and algorithm descriptions would be appreciated.

---

> ### Author Response · Authors · 2026-03-25
>
> Dear Reviewer JK3i,
>
> Thanks for your feedback and valuable comments. We reply to your comments as follows:
>
> **P1: "Demonstrating existence of CO in simplified models is valuable, but it does not necessarily provide full explanation of mechanism in general settings. Clarifying this distinction would strengthen contribution."**
>
> Thanks for highlighting this point. Let us clarify the distinction between our two main contributions.
>
> Our first contribution is showing that CO arises as a consequence of a phase transition. As you mention, this is mainly shown through two mathematical models (Section 3.1 and Appendix A.2). The primary goal of these mathematical models is to replicate the observations from large-scale models presenting CO, but in the smallest setting we can understand. The phase transition cannot be demonstrated in larger scales yet as included in our limitations (Section 7). However, we can: 1) Find a phase transition in two different architectures (Sinusoidal, Section 3.1 and ReLU, Appendix A2), while 2) Replicating empirical observations in the literature and 3) Extract insights (Corollary 4.3) that extrapolate to larger-scale models (Sections 6.1-6.2). These three points support the hypothesis that a phase transition induces CO.
>
> Our second contribution is showing that CO can occur with arbitrarily large $S$ and with random PGD initialization. Again, we resort to mathematically tractable models to demonstrate this phenomenon in small (Section 5.1) and large scales (Section 5.2). In this case, the main limitation is training dynamics. As stated in Section 7 (Limitations and future work), the practical manifestation of CO is also dictated by training dynamics. While we show that solutions with CO exist for any well-separated dataset, training algorithms might not converge to them. It is an interesting avenue to study CO from the training dynamics perspective.
>
> **P2: "Relatedly, CO can naturally occur when adversarial examples used during training differ from those used during evaluation. From this perspective, CO may be viewed as expected outcome rather than phenomenon requiring deeper explanation."**
>
> This is an interesting point. This perspective is aligned with robust overfitting [1]. Rice et al., [1] analyze the differences in performance under the same attack between the train and test sets, observing a slight degradation in the test set performance. Similarly, one could argue that a small degradation is expected when changing the attack used for evaluation. However, what makes CO catastrophic, is that the performance drops to nearly 0 [2,3]. This abrupt drop is what makes it interesting to understand the causes behind CO. In conclusion, yes, CO is somehow expected but its drastic effects have made it an interesting problem to study by its own.
>
>
> **P3: "Paper suggests that CO appears only above threshold $\epsilon_c$, yet discussion of why such threshold exists is limited.", **
>
> Thanks for highlighting this interesting point. Let us clarify our intuition behind the existence of $\epsilon_c$. Taking the sinusoidal model as an example, we can see that following Theorem 4.1, if the norm of our weight $|\theta|$ is unbounded, CO exists for any $\epsilon$. However, if the norm of the weight is constrained so that $|\theta| \leq B$, then, by Corollary 4.3, we have that the epsilon for which CO appears is $\epsilon \geq 2\pi S/ B$. This defines a clear critical value, as for $\epsilon \geq 2\pi S/ B$ we will find a specific solution presenting CO and for $\epsilon < 2\pi S/ B$ we will not. We will incorporate a discussion on these lines in the revised version of the manuscript.
>
> **P4: "it is not clear whether small training budget implies stability against CO, since training and testing attacks may use different budgets."**
>
> Interestingly, training with smaller epsilon provides stability. As studied by Andriushchenko and Flammarion [3], training with FGSM and $\epsilon\in \{5,6,7,8\}/255$ in CIFAR10 provides non-zero performance for even $\epsilon=16/255$, while training with $\epsilon=8/255$ provides 0 adversarial accuracy for all $\epsilon\geq 3/255$. These results are available in Figure 3 in [3]. We will discuss this result in the revised version of the manuscript.
>
> **P5: "In Algorithm 1, it is unclear whether index $i$ refers to individual samples or mini-batches."**
>
> As indicated in Line 1 of Algorithm 1, $M$ denotes the number of batches. Therefore, $i$ is the index of a batch. We will include a footnote clarifying that $\mathbf{x}_i$ and $y_i$ represent batches of inputs and labels in Algorithm 1.
>
> **P6: "In Section 3.1, cross-entropy loss is written as 𝓛(θ,ϵ), although it appears it should be written as 𝓛(θ)."**
>
> Thanks for highlighting this typo, indeed, the first appearance of $\mathcal{L}(\theta, \epsilon)$ in Section 3.1 should not depend on $\epsilon$. This will be fixed in the revised version of the manuscript.

---

> > ### Author Response · Authors · 2026-03-25
> >
> > **P7: "Equation (3) seems to require $\sum_i$"**
> >
> > Please not that since the loss is symmetric in this case and we are averaging across the two samples, the sumatory can be removed.
> >
> > **P8: "Corollaries 4.2 and 4.3 are labeled inconsistently in some places (e.g. Thm 4.2, Thm 4.3)."**
> >
> > Thanks for highlighting this typo. This is due to a missconfiguration of the cleveref package, it will be fixed in the revised version of the manuscript.
> >
> > **P9: "It would be helpful if authors could further explain how this (weight decay) relates to boundedness condition in Corollary 4.3."**
> >
> > The relationship between weight decay and the boundedness condition in Corollary 4.3 comes from an optimization point of view. Given a constrained optimization problem:
> > $$
> > \min_{\theta: |\theta|\leq B} f(\theta)
> > $$
> > An easier problem to solve is:
> > $$
> > \min_{\theta} f(\theta) + \lambda|\theta|
> > $$
> > for $\lambda>0$. Both problems share the same goal: make the norm of $\theta$ be low while minimizing the function $f$. The analysis in Corollary 4.3 follows the first case, while the weight decay experiments in section 6.3 follow the second case. The intuition is that if solving a problem like the first one can help avoid CO, solving a problem like the second one will also help avoid CO.
> >
> > We are thankful for your review and useful comments. We remain available to answer any follow-up questions appearing.
> >
> > Best regards,
> >
> > Authors
> >
> > **References**
> >
> > [1] Rice et al., Overfitting in adversarially robust deep learning. ICML 2020.
> >
> > [2] Wong et al., Fast is better than free: Revisiting adversarial training. ICLR 2020.
> >
> > [3] Andriushchenko and Flammarion, Understanding and Improving Fast Adversarial Training. NeurIPS 2020.

---

### Decision · Action_Editor_nFGf · 2026-04-17

**Recommendation:** Reject

**Audience:**

Yes

**Audience Explanation:**

The paper studies catastrophic overfitting in adversarial training, a topic of ongoing interest to researchers working on robustness, optimization, and the theoretical understanding of deep learning.

**Claims And Evidence:**

No

**Claims Explanation:**

The evidence provided is not sufficient to convincingly support the central claims. The theoretical analysis is primarily conducted in highly simplified or toy settings, and there is little empirical validation demonstrating that the findings extend to realistic models or practical adversarial training scenarios. Key claims regarding the underlying causes of catastrophic overfitting and the role of a critical adversarial budget are not adequately substantiated in practice.